# Significant influence of oxygenated volatile organic compounds on atmospheric chemistry: A case study in a typical industrial city in China

Jingwen Dai [a], Kun Zhang [a, *], Yanli Feng [a], Xin Yi [a], Rui Li [a], Jin Xue [a], Qing Li [a], Lishu Shi [a], Jiaqiang Liao [a], Yanan Yi [a], Fangting Wang [a], Liumei Yang [a], Hui Chen [a], Ling Huang [a], Jiani Tan [a], Yangjun Wang [a], Li Li [a, *]

[a] School of Environmental and Chemical Engineering, Shanghai University, Shanghai, 200444, China

*Correspondence*: Li Li (Lily@shu.edu.cn) and Kun Zhang (zk1231@shu.edu.cn)

## Abstract

Oxygenated volatile organic compounds (OVOCs), an important subgroup of volatile organic compounds (VOCs), are emitted directly or formed secondarily through photochemical processes. They play a crucial role in tropospheric chemistry as ozone ($O_3$) precursors. However, due to measurement limitations, the influence of OVOCs on $O_3$ formation has often been underestimated. In this study, 74 VOCs (including 18 OVOCs) were measured at five representative stations (urban, suburban, industrial, upwind, and downwind stations) in Zibo, an industrial city in the North China Plain. The VOCs level in Zibo ($44.6 \pm 20.9 \times 10^{-9}$) is in the upper-middle range ($> 32 \times 10^{-9}$) compared to previous studies conducted in most Chines cities, with OVOCs contributing for 30.0%~37.8%. The average $O_3$ formation potential in Zibo is $410.4 \pm 197.2 \ \mu g \ m^{-3}$, with OVOCs being the dominant contributor (31.5%~55.9%). An observation-based model (OBM) was used to access the contributions of chemical production ($R_{NetProd}$) and emissions/transport ($R_{Emis\&Trans}$) to individual OVOC. Daytime (8:00-18:00 LT) $R_{NetProd}$ is the highest at the urban site ($5.9 \times 10^{-9} \ h^{-1}$), while nighttime $R_{Emis\&Trans}$ is most negative at the industrial site ($0.76 \times 10^{-9} \ h^{-1}$). Simulations without

OVOCs constraint overestimate OVOCs (42.1~126.5%) and key free radicals (e.g., hydroperoxy

radicals ($HO_2$, 5.3%~20.4% and organic peroxy radicals ($RO_2$, 6.6%~35.1%)), leading to a

1.8%~11.9% $O_3$ overestimation. This overestimation causes an underestimation of hydroxyl radicals

(OH) (1.8%~20.9%) and atmospheric oxidizing capacity (3.5%~12.5%). These findings emphasize

the importance of comprehensive OVOCs measurements to constrain numerical models, especially

in regions with dense anthropogenic emissions, to better reproduce atmospheric photochemistry,

and to formulate more effective air pollution control strategies.

## 1. Introduction

Oxygenated volatile organic compounds (OVOCs), contributing 20.1%~73.5% of total volatile

organic compounds (VOCs) (Han et al., 2019; Huang et al., 2020; Li et al., 2022a; Liu et al., 2024;

Song et al., 2024), are critical components of tropospheric photochemistry (Yang et al., 2014).

Photolysis of OVOCs has been proved to be the most significant primary source of $RO_x$ (OH + $HO_2$

+ $RO_2$) in Guangzhou, Beijing, and Xi'an in China (Wang et al., 2022c; Yang et al., 2018; Zhang et

al., 2021b), and thereby accelerating the recycling of radicals to promote ozone ($O_3$) formation (Qu

et al., 2021; Wang et al., 2022c). In addition, previous studies have shown that sufficient free radicals

produced by photolysis of OVOCs are the dominated contributors to $O_3$ pollution during winter

(Edwards et al., 2014; Emmerson et al., 2005). The study of Li et al. (2021b) indicates that the fast

generation of $O_3$ during winter haze in the North China Plain is mainly driven by the photolysis of

formaldehyde (HCHO), which leads to a large production of $HO_x$ radical and offsets the radical

titration induced by $NO_x$ emissions. In addition, HCHO and other OVOCs dominated the OH loss

with VOCs (Goldan et al., 2004), resulting in predominant role in OH reactivity (Ling et al., 2014;

Yang et al., 2018). Therefore, OVOCs play a significant role in the atmospheric chemistry.

OVOCs have complex and diverse sources, including primary emissions from anthropogenic,
e.g., vehicle exhausts (Gentner et al., 2013; Legreid et al., 2007; Wang et al., 2022b), volatile
chemical product use (Ou et al., 2015), industries (Wang et al., 2023), biomass combustion (Gilman
et al., 2015; Karl et al., 2007; Li et al., 2014a; Yokelson et al., 2007), and biogenic sources (Ou et
al., 2015; Rieksta et al., 2023). They are also formed secondarily through photochemical reactions
(Huang et al., 2020; Song et al., 2024; Xia et al., 2021). Mo et al. (2016) estimated that OVOCs
from heavy-duty diesel vehicle emissions accounted for 53.8% of total VOCs in China, and OVOCs
account for 12.4%~46.3% of VOCs emission from biomass and residential coal combustion, which
demonstrates the importance of combustion-related sources of OVOCs. In addition, measurement
of VOCs fluxes based on the airborne eddy covariance technique showed that urban emission
sources comprise a surprisingly large proportion of OVOCs (29%~56%) (Karl et al., 2018;
Pfannerstill et al., 2023). Due to the high share of OVOCs in VOCs, previous studies have reported
that OVOCs could contributed 38%~60% of ozone formation potential (OFP) (Liu et al., 2024; Mo
et al., 2022; Wang et al., 2022a, 2024). The loss of OVOCs occurs through photolysis, reactions
with oxidants (e.g., OH, $NO_3$, and $O_3$), dilution mixing and deposition (Atkinson, 2000; Atkinson
and Arey, 2003). Moreover, air mass transport also can significantly affect the mixing ratio of
OVOCs.
Chemical transport models (CTMs) have been widely used for the study of formation
mechanism of OVOCs and their influence on air quality (Chen et al., 2022; de Gouw et al., 2018;
Luecken et al., 2012; Steiner et al., 2008; Yang et al., 2023). However, due the deviation of the
meteorological field, uncertainty of the emission inventory (Li et al., 2019; McDonald et al., 2018;
Shen et al., 2019), defects of lumped chemical mechanism (Li et al., 2014b; Sarwar et al., 2008;
Stockwell et al., 1997a; Venecek et al., 2018), there is a large uncertainty in the OVOCs simulated
by CTMs, which in turn leads to large deviations in the simulated atmospheric photochemistry. The
observation-based model (OBM) can avoid these biases to a certain extent by constraining
meteorological parameters and chemical species, and leveraging detailed chemical mechanism (e.g.,
Master chemical mechanism, MCM). Nevertheless, due to the limited observations of OVOCs (e.g.,
Pfannerstill et al., 2023), many existing studies use OBM without the observed OVOCs data, or
only with limited inputs for certain OVOCs species (formaldehyde, acetaldehyde, acetone), which
can greatly bias the assessment of $O_3$ generation mechanism, free radical chemistry, and atmospheric
oxidation. Wang et al. (2022a) showed that the box model without the constraint of OVOCs
underestimates the OVOCs concentrations, which in turn lead to the underestimation of $RO_x$ and $O_3$
formation. Thus, it is meaningful to couple OVOCs observation with OBM to investigate how
OVOCs affect radical chemistry, atmospheric oxidization capability, and $O_3$ formation mechanism.

Zibo, a typical industrial cluster city in China, has been suffering from $O_3$ pollution for years

(Li et al., 2021a; Qin et al., 2023). However, comprehensive studies involving the observation of
VOCs, particularly OVOCs such as HCHO, are rare. Qin et al. (2023) used observations of 98 VOCs
(without HCHO) in Zibo to constrain OBM, but the absence of HCHO in their simulation could
result in underestimation $RO_x$, thus disturbing the investigation of OH budget (Fuchs et al., 2017;
Guo et al., 2021; Ling et al., 2014; Qu et al., 2021; Tan et al., 2017). This study hypothesizes that
incorporating observational constraints on OVOCs significantly influences the OBM simulations.
To evaluate this, a 5-day field campaign was conducted across five representative sites in Zibo.
Concentrations of 74 VOC species, including 29 alkanes, 16 aromatics, 9 alkenes, 18 OVOCs,
acetylene and isoprene, are obtained. The contributions of secondary formation, emissions/transport
to OVOCs level are analyzed by the OBM. Additionally, the impact of OVOCs on radical chemistry,
atmospheric oxidation capability, and consequently $O_3$ production are quantified.

## 2. Methodology

### 2.1 Sampling sites and measurements

To capture a typical ozone pollution case, the field campaign was conducted from August 8 to

August 12, 2021, at five monitoring sites (Zhonglou (ZL), Chengdong (CD), Chengqu (CQ),
Tianzhen (TZ), and Xindian (XD)) in Zibo (Figure 1, Table S1). Among the five sites, ZL site
(117°54'E, 36°39'N) is an urban site, which is located in the central area of Zibo, and is mainly
surrounded by residential areas and factory buildings. According to the prevailing wind direction
(northeast, Figure S1 (a)), CQ (118°60'E, 36°57'N) site is an upwind site, while CD (117°53'E,
36°31'N) is a downwind site. CD is located on a hillside in the southern part of Zibo, with a small
number of ceramic and refractory enterprises factories nearby. TZ (117°48'E, 37°10'N) is close to
Shengli Oil field on the west, and is surrounded by farmland. This site is regarded as a suburban site
and is affected by residential emissions in the north of Zibo, as well as nearby oil production
operations. XD (118°19'E, 36°48'N) is close to a chemical industrial park and serves as an industrial
site. More detailed information about these sites can be found in Table S1.

Site-scale wind patterns can affect the levels and spatial distribution of OVOCs and PAMS

(target VOC species from the Photochemical Assessment Monitoring Stations, including include 29
alkanes, 16 aromatics, 9 alkenes, isoprene, and acetylene) across sites. Urban (ZL) and downwind
(CD) sites are impacted by OVOC pollution from northeasterly (NNE, NE, ENE) winds, while the
upwind (CQ) site experiences higher OVOC and VOCs pollution under both northeasterly and
northwesterly (WNW, NW, NNW) winds (Figure S2 (a, b)). Suburban (TZ) and industrial (XD)
sites exhibited higher OVOC and VOCs levels under southeasterly (SE) and southwesterly (WSW)
winds, respectively, likely due to upwind emissions from nearby industrial sources. WS between 1
and 2 m s$^{-1}$ were most common (40.4%) during the observation period. At suburban (TZ) and
industrial (XD) sites, OVOC and VOCs levels were lower at low wind speeds (WS < 2 m s$^{-1}$) than
that at higher wind speeds, reflecting the influence of local emissions (Figure S2(c, d)). In contrast,
at downwind (CD) site, higher OVOC and VOCs levels were observed at WS > 2 m s$^{-1}$, indicating
the impact of regional transport. At the urban (ZL) site, higher WS are associated with lower VOCs
levels and higher OVOC levels, indicating the influence of aging air masses transported from
upwind regions.

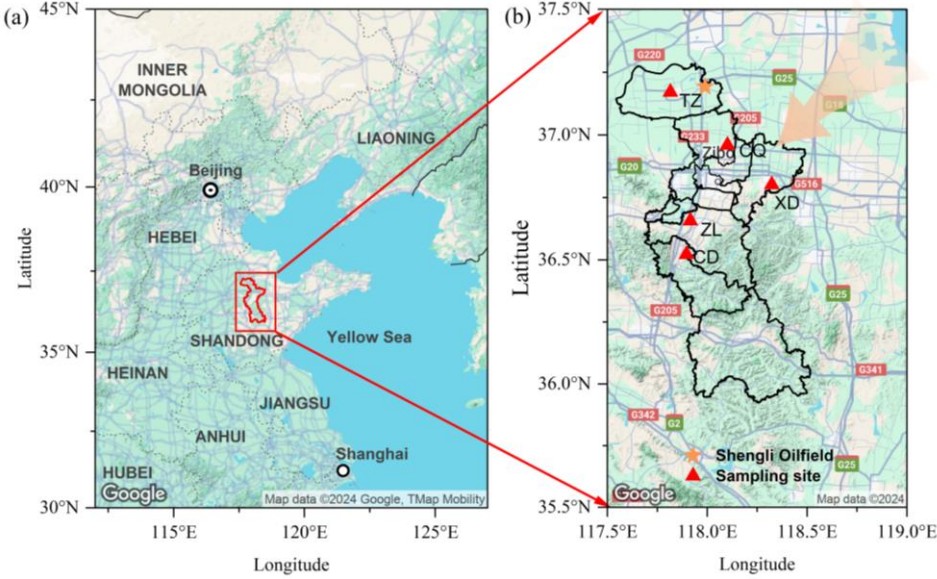


**Figure 1. (a) Map of the North China Plain and (b) map of Zibo with locations of VOCs monitoring stations**
**(red triangles), and Shengli Oil field (yellow star), and prevailing wind direction (orange arrow).**

During the campaign, two online gas chromatography-flame ionization detector (GC-FID,

Thermo Scientific GC5900) systems were deployed at suburban (TZ) and upwind (CQ) stations,
and three online gas chromatography-flame ionization detector (GC-FID, Syntech Spectras GC
955–615/815) systems were deployed at downwind (CD), industrial (XD) and urban (ZL) stations,
respectively. A mixture of 56 PAMS target species (Spectra Gases Inc., USA, Table S2) was used
for the calibration of the GC-FID system. Each VOC analyzer provided measurements with a 1-
hour temporal resolution. More detailed descriptions of these instruments can be found in previous
studies (Li et al., 2023; Wang et al., 2014; Yang et al., 2022; Zheng et al., 2023). Overall, the
detection limits for most VOC species are below $0.1 \times 10^{-9}$. Additionally, 18 oxygenated VOCs
species were collected by 2,4-dinitrophenylhydrazine (DNPH) sorbent tubes in conjunction with an
automated sampler for a period of 1 or 3 hour per sample. A 47 mm quartz filter membrane is
attached to the front of the sampling tube to filter particulate matter. An ozone scrubber (silica gel
column tubing coated with potassium iodide) was placed at the front of the air inlet to avoid ozone
interference. OVOCs were derivatized in cartridges to hydrazones during sampling. The cartridges
were eluted with 3 mL of acetonitrile and stored at 0-4 °C immediately. Then the eluants were
analyzed using an Agilent HPLC, equipped with ultraviolet absorption detector (UVD), quadruple
pump, and Agilent TM C18 reversed column (250 mm×4.6 mm, 5.0 μm). A gradient elution was
used, and the mobile phase was mixing of acetonitrile, tetrahydrofuran and water. The analysis was
carried out using a ternary gradient elution program at a flow rate of 1.2 mL/min, with detection
wavelength of 360 nm, and sample volume of 10 μL at a column temperature of 45 °C. More details
about OVOC samplings and analysis can be found in Peng et al. (2023). The lower limit of detection
for OVOCs were $<0.1 \times 10^{-9}$ (Peng et al., 2023). A total of 271 valid OVOCs samples were collected
during the campaign. At the industrial (XD) and suburban (TZ) stations, 8 samples were collected
per day at 3-hour intervals. At the urban (ZL), upwind (CQ) and downwind (CD) stations, 10
samples were collected per day, with 7 samples collected at 1-hour intervals during 7:00-21:00 LT,
and 3 samples collected at 3-hour intervals during the night (1:00-6:00 and 22:00-1:00[+1] LT), and
totaling 59 valid samples per station. Finally, a total of 74 VOCs (56 PAMS and 18 OVOCs) were
combined to conduct data analysis in this study (Table S2). Conventional gas phase pollutants (e.g.,
$O_3$, CO, and $NO_x$ (NO and $NO_2$)) were measured using commercial online analyzers (Thermo
Scientific 49i, 48i, and 42i, USA) at each site. $NO_x$ was measured by NO chemiluminescence and
chemical conversion with a molybdenum convertor, which is known to be interfered by $NO_z$ species
(Tan et al., 2017, 2019a). Meteorological parameters, including temperature (T), relative humidity
(RH), wind speed (WS), wind direction (WD), and ambient pressure (P) were obtained
synchronously by Chinese ground-based meteorological stations (Boshan, Huantai, Gaoqing, Linzi,
and Zichuan sites) (http://data.cma.cn/, last access: March 26, 2024).
**2.2 Observation-based model**

A box model (F0AM) coupled with the Master Chemical Mechanism (MCM) v3.3.1 was

utilized to simulate the in situ atmospheric chemical process at these 5 sites (Jenkin et al., 2015;
Wolfe et al., 2016). The MCMv3.3.1, as a nearly explicit mechanism with more than 5800 species
and 17000 reactions, provides a more detailed gas chemistry than other lumped mechanisms, such
as the Carbon Bond Mechanism (CB) (Yarwood et al., 2005, 2010), Regional Atmospheric
Chemistry Mechanism (RACM) (Goliff et al., 2013; Stockwell et al., 1997b), and SAPRC (Carter,
1990, 2010b; Carter and Heo, 2013). The box model calculations were constrained by
comprehensive measurements of trace gases (NO, $NO_2$, CO, and $SO_2$) and 45 speciated VOCs,
encompassing 20 alkanes, 9 alkenes, 14 aromatics, 15 OVOCs, isoprene and acetylene, as well as
meteorological parameters (T, RH and P). To address potential $NO_2$ measurement artifacts, several
adjustments were implemented. Considering that PKU-Mo as a catalytic converter for $NO_2$
measurement can cause interferences from other nitrogen–oxygen compounds (e.g., PAN, $HNO_3$),
potentially overestimating $NO_2$ by 30%~50% (Kim et al., 2015; Tan et al., 2017, 2019a; Xu et al.,
2013). In this study, the observed $NO_2$ concentrations at the 5 sites were reduced by 30%~40% (40%
for ZL and CQ, 30% for CD, TZ and XD) to compensate for catalytic converter interferences (Xu
et al., 2013). Additionally, strong anthropogenic emissions (e.g., vehicle emissions) near the sites
may prevent the model from reaching steady state, leading to positive deviations (Li et al., 2014c).
Therefore, NO steady-state approximations ($NO_{ss}$), calculated according to the equations proposed
by Del Negro et al. (1999) (Equation S1), was used to constrain the simulated NO. The uncertainties
derived from the $NO_x$ settings are shown in Table S3 and analyzed in Section 3.4. HONO was fixed
to 2% of the corrected $NO_2$ mixing ratio (Elshorbany et al., 2012; Tan et al., 2019a), and the
corresponding uncertainty is summarized in Section 3.4. In addition, boundary layer height (BLH),
and surface net solar radiation (SSR) were obtained from the fifth generation of the European Centre
for Medium-Range Weather Forecasts (ECMWF) reanalysis for the global climate and weather
(https://cds.climate.copernicus.eu, last access: March 1, 2024). The photolysis frequency correction
factor (Jcorr) of the model input was adjusted by SSR. BLH was also included in the model to
control the deposition process (Xuan et al., 2023; Zhu et al., 2020).

The model ran with continuous time series profile for the campaign period (August 8-12) with

1-hour time-step. A sensitivity analysis was performed for the time-step and the results are
summarized in Section 3.4. Each simulation started with 10-days spin up to reach steady state
condition. Missing observation data were filled with linear interpolation, and the mixing ratios of
OVOCs were also linearly interpolated to 1-hour resolution for modeling. An artificial loss process
corresponding to an atmospheric lifetime of 24 h or a first-order dilution rate (kdil) of $1/86400$ $s^{-1}$
was introduced for all simulated species, including secondary species and radicals, to approximately
simulate dry deposition and other losses (Lou et al., 2010; Tan et al., 2018b; Wang et al., 2022c).
The model cases that run with the above settings with 15 constrained OVOCs species are called the
Base scenario. To investigate the impacts of constrains of OVOCs on atmospheric chemistry, the
Free scenario was conducted, with all the setting of the Base scenario except for the OVOCs
constraint.

**2.3 Budgets of OVOCs and O₃**

At a given site, variations in OVOCs mixing ratios are mainly influenced by in-situ

photochemical production and chemical loss, emissions, regional transport, and deposition (Tan et
al., 2018a; Xue et al., 2014a; Zhang et al., 2021). The change rate of observed OVOCs ($R_{Meas}$) is
calculated by Equation (1). The in-situ photochemical production of OVOC ($R_{ChemProd}$) is mainly
caused by the oxidation of VOCs, while their in-situ chemical loss ($R_{ChemLoss}$) includes photolysis
and reactions with oxidants (OH, $NO_3$, and $O_3$) (https://mcm.york.ac.uk/MCM/, last access: 13 Jan
2025) (Atkinson, 2000; Atkinson and Arey, 2003; Jenkin et al., 2015; Saunders et al., 2003). The in-
situ net OVOCs chemical production ($R_{NetProd}$) (Equation (2)) and their removal by deposition ($R_{Deps}$)
are calculated hourly according to the OBM simulation. The OBM primarily accounts for
atmospheric photochemical reactions, and deposition within the boundary layer. However, previous
studies have reported that the OBM lacks an explicit representation of transport processes and
emissions (Wolfe et al., 2016; Zhang et al., 2021), making it challenging to disentangle their
respective contributions. Therefore, emissions and transport are combined to a single term
($R_{Emis\&Trans}$) to represent their contributions collectively. If the $R_{Emis\&Trans}$ is positive, it is considered
a net import of emissions/transport, whereas a negative suggests a net export. The emissions and
regional transport of OVOCs ($R_{Emis\&Trans}$) are computed as Equation (3).

$$R_{Meas} = \sum_i \frac{d([OVOC]_i)}{dt} \tag{1}$$

$$R_{NetProd} = \sum_i (R_{ChemProd,i} - R_{ChemLoss,i}) \qquad (2)$$

$$R_{Emis\&Trans} = (R_{Meas} - R_{NetProd} - \sum_i R_{Deps,i}) \qquad (3)$$

where $[OVOC]_i$ is the mixing ratios of OVOC species $i$ constrained in OBM, 15 in total (Table S2).
dt is the time-step of modeling, $d[OVOC]_i$ refer to the change in mixing ratio of OVOC species $i$.

Considering the oxidation of NO to $NO_2$ by peroxyl radicals, the total oxidant ($O_x = O_3 + NO_2$)

is generally used to characterize the chemical budget of $O_3$ (Kanaya et al., 2009; Xue et al., 2014b).
The total chemical production of $O_x$ through oxidations of NO by $HO_2$ and $RO_2$ radicals (Tan et al.,
2018b), is defined as the production of $O_3$ ($P(O_3)$), which is calculated according to Equation (4):

$$P(O_3) = k_{HO2+NO}[HO_2][NO] + \sum k_{RO2,j+NO}[RO_2]_j[NO] \qquad (4)$$

The chemical loss rate ($L(O_3)$) of $O_3$ is equal to the sum of loss rates of $O_3$ and $NO_2$, including

$O_3$ photolysis, reactions of $O_3$ with OH, $HO_2$ and alkenes, as well as reactions of $NO_2$ with OH and
$RO_2$, as well as the reaction of $NO_3$ with unsaturated VOCs (Chen et al., 2020; Liu et al., 2022; Xue
et al., 2016, 2014b).

$$L(O_3) = k_{O1D+H2O}[O1D][H_2O] + k_{O3+OH}[O_3][OH]$$
$$+ k_{O3+HO2}[O_3][HO_2] + k_{O3+alkenes}[O_3][alkenes]$$
$$+ k_{NO2+OH}[NO_2][OH] \qquad (5)$$
$$+ \sum k_{NO2+RO2,j}[NO_2][RO_2]_j$$
$$+ \sum k_{NO3+VOC,i}[NO_3][VOC]_i$$

The concentrations of radicals and intermediates are derived from the outputs of the OBM. The

$k$ values in Equations (4) and (5) rate constants of the corresponding reactions, which can be found
from https://mcm.york.ac.uk/MCM/ (last access: 13 Jan 2025) or the study by Liu et al. (2022). The
subscript '$j$' in Equation (4) and (5) denotes individual $RO_2$ species. The subscript '$i$' in Equation
(5) represents individual VOC species. The net $O_3$ production rate can be obtained from the
difference between P ($O_3$) and L($O_3$).
**2.4 Evaluation of ozone formation potential and atmospheric oxidation capacity**
Different VOC species vary in their capability to form ozone, and their potential to produce $O_3$
can be evaluated by the maximum incremental reactivity (MIR) (Carter, 2010a). The ozone
formation potential (OFP) calculated for each VOC species represents its maximum contribution to
ozone production (Bufalini and Dodge, 1983). The OFP of VOCs is calculated as follows:

$$OFP_i = [VOC]_i \times MIR_i \tag{6}$$

where $OFP_i$ is the OFP of VOC species $i$ ($\mu$g m$^{-3}$), $[VOC]_i$ is the atmospheric concentration of VOC
species $i$ ($\mu$g m$^{-3}$), $MIR_i$ is the maximum incremental reactivity coefficient of the VOC species $i$ (g
$O_3$/g VOCs) (Table S2) from Carter, 2010a.
Atmospheric oxidation capacity (AOC) is the core driving force of complex air pollution,
influencing the removal rate of trace gases and the production rates of secondary pollutants (Liu et
al., 2021). AOC is calculated based on the sum of oxidation rates of oxidants (OH, $O_3$, and $NO_3$)
with primary pollutants (VOCs, CO, and $CH_4$) (Elshorbany et al., 2009; Geyer et al., 2001; Yang et
al., 2022b). The formula is as follows:

$$AOC = \sum_i k_{Yi}[Y_i][X] \tag{7}$$

where $Y_i$ represents primary VOCs (excluding OVOCs), CO and $CH_4$, $X$ represents oxidants (OH,
$O_3$ and $NO_3$) and $k_{Yi}$ is the bimolecular rate constant for the reaction of $Y_i$ with $X$. Atmospheric
oxidation capacity determines the rate of $Y_i$ removal.
**3. Results and discussion**
**3.1 Meteorological and chemical conditions**

The field campaign is characterized by consistent hot and sunny conditions, with the average

daily maximum temperature and SSR of 32.2±1.4 °C (peak at 34.1 °C) and 2.1±0.4× $10^6$ J $m^{-2}$
(Figure 2, Figure S2 (a)), respectively, which favors the photochemical formation of $O_3$. A typical
$O_3$ episode was observed, with an average maximum daily 8-hour average $O_3$ (MDA8-$O_3$) of 89.8
× $10^{-9}$ in Zibo city. According to the Chinese National Ambient Air Quality Standard Grade II (about
93.3 × $10^{-9}$ for 1-hour average, or 74.7 × $10^{-9}$ for MDA8-$O_3$), there are four $O_3$ pollution days
(August 8 to 11) during the campaign. The average mixing ratios of $SO_2$, $NO_2$, NO, and CO in Zibo
are 2.8±1.6, 12.0±6.9, 2.8±4.8, and 897±670 × $10^{-9}$, respectively (Figure 2, Figure S3 (b)). The mean
VOCs mixing ratio in this study is 44.6±20.9 × $10^{-9}$, which is overall higher than that in Beijing
(18.3±8.9 × $10^{-9}$) from July 23 to August 31 in 2016 (Wu et al., 2023), Rizhao (9.83 × $10^{-9}$) in
summer in 2022 (Zhang et al., 2023), and Xi'an (29.1± 8.4 × $10^{-9}$) from June 20 to July 2019 (Song
et al., 2021). Compared with the median VOC levels (~32 × $10^{-9}$) in other cities in China (Figure 3,
Table S4), VOC levels in Zibo is in the upper-middle range. Previous studies have demonstrated
that industrial processes account for approximately 49% of total VOC emissions in Shandong
Province (Jiang et al., 2020; Li et al., 2017; Ren, 2011; Zheng et al., 2021). This indicates strong
anthropogenic VOCs emission in Zibo. Notably, VOC emission intensity in Zibo was among the
highest in Shandong Province, with values > 90 t $km^{-2}$ $y^{-1}$, even >108 t $km^{-2}$ $y^{-1}$in some areas in
2016 (Jiang et al., 2020; Zhou et al., 2021). In terms of the VOCs groups (Figure 5 (c), Figure S4
(a)), alkanes and OVOCs were the two predominant groups at each site, accounting for 33.3~51.5%
and 30.0~37.8% to the total VOCs, respectively, followed by aromatics (3.8~16.5%) and alkenes
(5.0~13.8%). In addition, the difference between peak and valley $NO_2$ mixing ratios was 14.4±3.2
× $10^{-9}$, indicating that substantial $NO_x$ was converted to $O_3$.

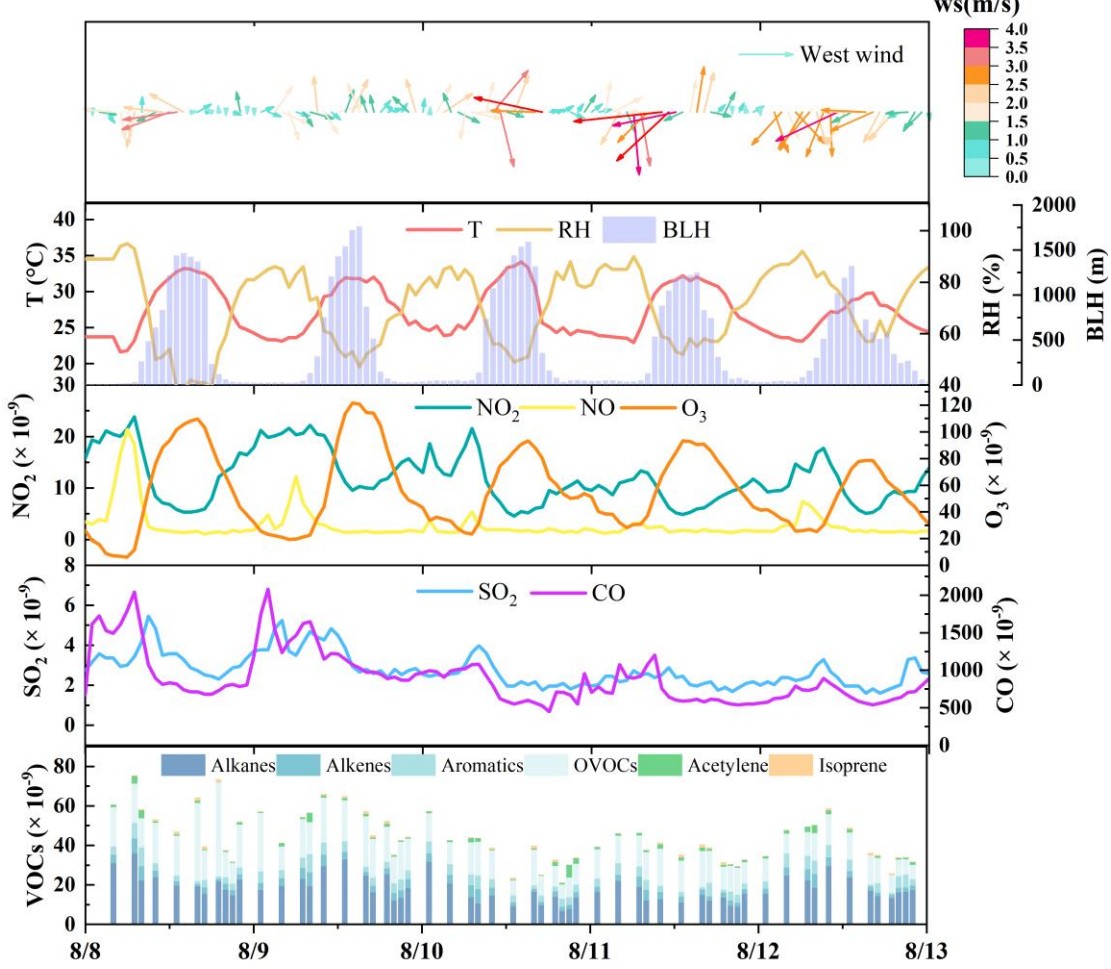


**Figure 2. Time profiles of pollutant mixing ratios and meteorological parameters in Zibo from August 8 to**
**12, 2021. The meteorological data were from ZL, the central site of Zibo, and the pollutants data were the**
**average of the five sites. The hourly PAMS (including alkanes, alkenes, aromatics, acetylene, and isoprene)**
**data were aligned with the 1/3-hour sampling intervals of the OVOCs data to ensure comparability between**
**the two datasets.**

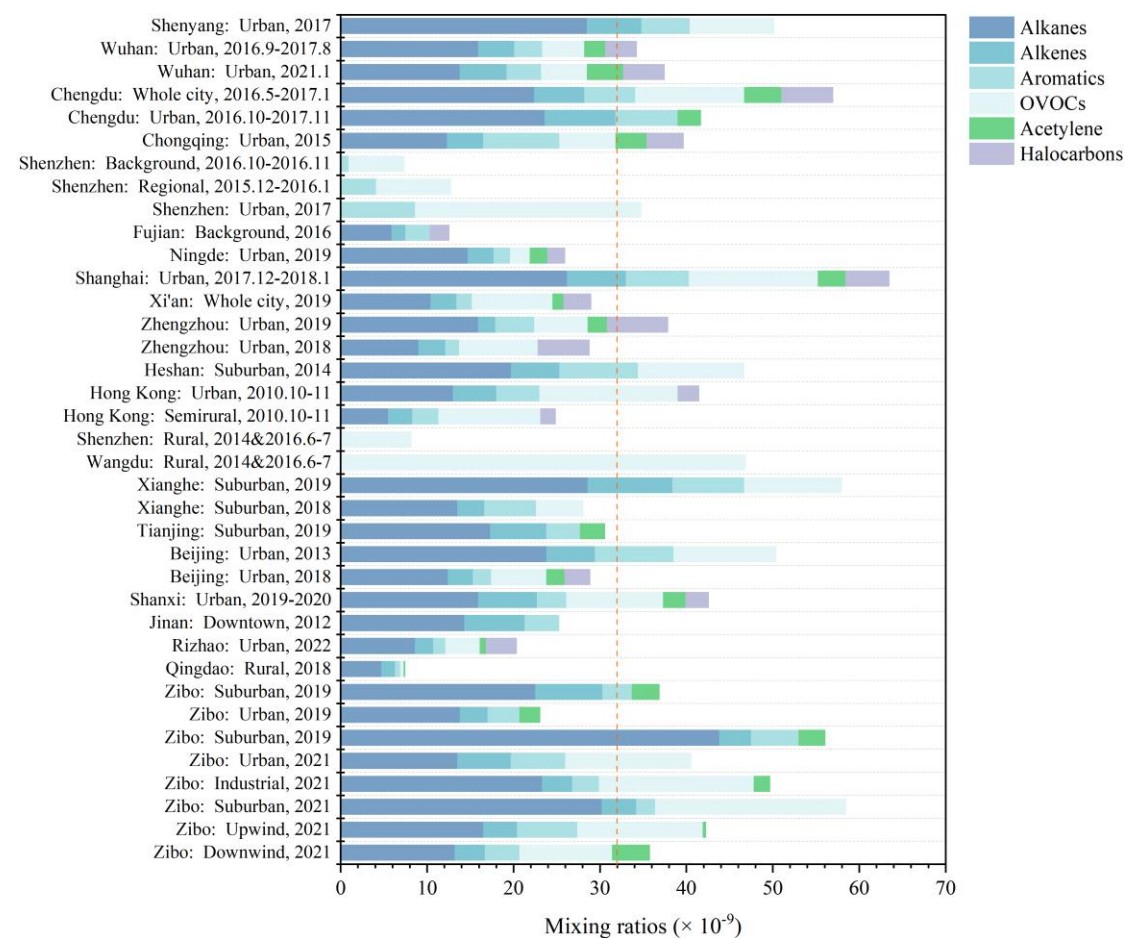


**Figure 3. Comparison of VOCs mixing ratios and compositions in this study with former studies based**
**on Table S4. The red dash line represents the median levels (~32 × 10⁻⁹) of VOCs。**

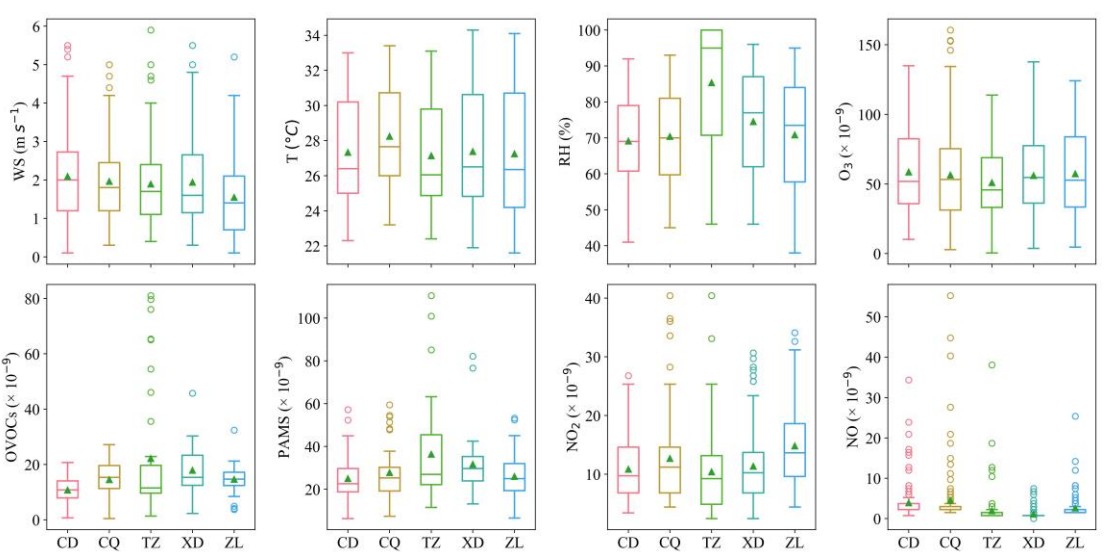


**Figure 4. Distribution of O₃ and its precursors (OVOCs, PAMS, NO₂, and NO) and meteorological**
**parameters (WS, T, and RH) at five sites, with the green triangle being the mean value and the horizontal**
**line in the box being the median.**
Across the five sites, the O₃ mixing ratios are comparable across all the sites (Table S6, Table
S11) (p > 0.05). However, TVOCs at suburban site (TZ, $58.5\pm35.0 \times 10^{-9}$) is the highest (Figure 5
(c), Figure S4 (b)), which is attributed to oil refineries near this site. The downwind site (CD) has
slightly lower $NO_2$ level ($10.8\pm5.1 \times 10^{-9}$) and lower TVOCs mixing ratios ($35.7\pm12.5 \times 10^{-9}$) than
urban site (ZL, $14.8\pm6.5$ and $40.6\pm10.3 \times 10^{-9}$, respectively) and upwind site (CQ, $12.7\pm8.1$ and
$42.3\pm15.4 \times 10^{-9}$, respectively), and has higher $O_3$ mixing ratio ($58.6\pm30.0 \times 10^{-9}$) than CQ and ZL
station. This may be attributed to the sequential transport of $O_3$ and its precursors from the upwind
station (CQ) to urban station (ZL), and subsequently to the downwind station (CQ), driven by the
dominant northeasterly winds (Figure 1 (b), Figure S1).

According to the time series of individual pollutant (Figure S3 (b)), CQ showed obvious peak

mixing ratios of $O_3$, $NO_2$, NO and CO than the other sites during August 8-9, with stagnant
conditions (WS < 2 m s$^{-1}$), indicating stronger emissions from combustion sources and possibly fast
photochemical process near CQ. In addition, XD showed high mixing ratios of CO during August
8-9, and high daytime TVOCs levels on August 9 (9:00-14:00 LT, $90\sim110 \times 10^{-9}$). Given CO's
relatively inert nature and the absence of similar CO peaks at the other four sites, the abnormal CO
peak at XD is related to strong emissions from nearby factories in the industrial park. TZ showed
distinct morning and evening peaks of TVOCs at 6:00 LT ($163.0 \times 10^{-9}$) and 21:00 LT ($120.0 \times 10^{-9}$)
on August 8, and a night peak at 1:00 LT on August 10 ($130.3 \times 10^{-9}$), which were attributed to
emissions from the neighboring oil field operations. From August 10 to 12, as wind speeds increased,
pollutants levels at all sites decreased to similar levels. Overall, local anthropogenic emissions in
Zibo were more prominent under weak wind conditions.

To compare the secondary $O_3$ formation in each site, the ozone formation potential (OFP) of

each VOCs is calculated (Equation (6)). The mean OFP in Zibo during the observation is
$410.4\pm197.2$ µg m$^{-3}$, with OVOCs accounting for the largest proportion ($31.5\%\sim55.9\%$), followed
by aromatics ($10.2\%\sim41.2\%$). Alkanes ($10.3\%–24.6\%$) and alkenes ($11.4\%\sim23.1\%$) make
comparable proportions, while BVOCs accounted for only $2.1\%\sim7.6\%$ of the total OFP (Figure 5
(b)). The one-way analysis of variance (ANOVA) results ($p < 0.05$) indicate significant differences
(Armstrong et al., 2000) in VOC subclass contribution to OFP across the 5 sites. Alkanes and
aromatics show larger F-values (Table S11), reflecting greater variations in the contributions to OFP
across the 5 sites, whereas OVOC and BVOC (isoprene) exhibited lower variability. Post-hoc Tukey
honestly significant difference (HSD) tests were performed followed ANOVA to further identify
specific significant differences in VOC subcategories between sites (Figure S14). The OFP of
TVOCs is generally similar across stations, except for the downwind station (CD). The upwind
station (CQ, $464.2\pm162.3$ µg m$^{-3}$) has the highest OFP, followed by the suburban site (TZ,
$456.3\pm295.3$ µg m$^{-3}$), the urban site (ZL, $441.1\pm174.5$ µg m$^{-3}$), the industrial site (XD, $422.9\pm166.9$
µg m$^{-3}$), and the downwind site (CD, $279.4\pm101.2$ µg m$^{-3}$) (Table S5). Differences in OFP levels of
aromatics and alkanes at downwind station (CD), suburban station (TZ) and industrial station (XD)
are minimal (Figure S14 (b, c)). However, significant differences in OFP contributed by OVOC at
downwind station (CD) compared to suburban station (TZ) and industrial station (XD) are attributed
to OVOC emission sources, regional transport and secondary formation (Figure S14 (a)). Apart from
CQ, OVOCs are the dominant contributors to OFP at each site, especially TZ and XD, with mean
OFP of $254.9\pm276.1$ µg m$^{-3}$ ($55.9\%$) and $194.7\pm101.0$ µg m$^{-3}$ ($46.0\%$) from OVOCs, respectively.
This indicates the key role of OVOCs in the formation of O$_3$ at our observational sites. Among
OVOC species, HCHO is the dominant contributor to OFP across the five sites ($56.6\sim202.0$ µg m$^{-3}$
$^{3}$). This is consistent with previous studies (Duan et al., 2008; Huang et al., 2020; Zhou et al., 2024).
The top four OVOC species are formaldehyde, acetaldehyde, propionaldehyde, and butyraldehyde,
which cumulatively contributed 91%~95% of the OFP from OVOCs (Table S5). Additionally, the
variety of VOCs sources, meteorological condition, and photochemical condition in each site lead
to differences in key species of OFP at each site. At the suburban site (TZ), isoprene (34.9 μg m$^{-3}$)
ranks 2$^{nd}$ in terms of OFP after formaldehyde (202.0 μg m$^{-3}$), indicating a high impact of biogenic
emissions (Mo et al., 2018; Sindelarova et al., 2014). At the industrial site (XD), the contribution of
isopentane, marker of oil and gas emissions, to OFP is more prominent (as high as 66.2 μg m$^{-3}$) than
other sites. OFP of highly reactive aromatic hydrocarbon species, such as m/p- xylene (53.8 μg m$^{-3}$
$^{3}$) and o-xylene (23.6 μg m$^{-3}$) are predominant at upwind site (CQ), indicating outstanding
contribution of solvent-using sources. OFP contributed by alkenes is the highest at urban site
(101.8±56.8 μg m$^{-3}$) (Figure 5 (a)), with ethylene and propylene being the most key species, which
is consistent with the dense vehicle emission near this site.

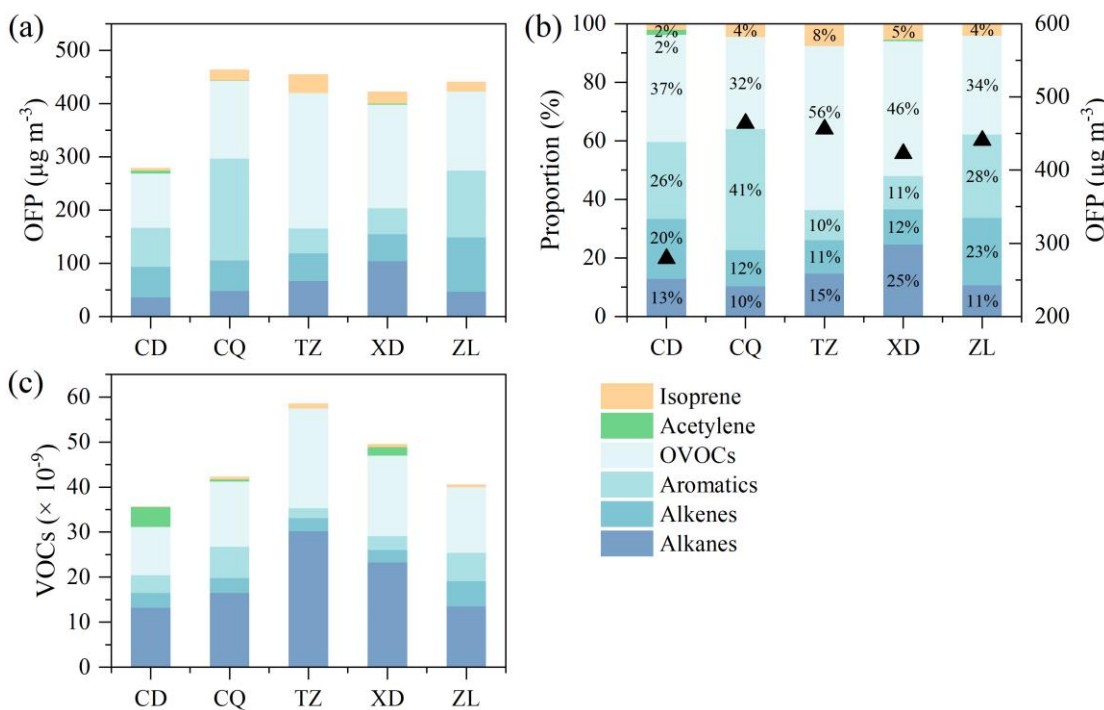


**Figure 5. (a) Ozone formation potential (OFP) and (b) proportions of OFP contributed by VOCs subgroups,**
**along with (c) mixing ratios of VOCs subgroups at five sites.**

**3.2 Contribution of chemical generation, emissions/transport to OVOCs**

OBM simulation results were used to analyze the contributions of chemical processes, and emissions/transport to OVOCs. Overall, the modeled $O_3$ in the Base scenario exhibited good model performance at the five sites, with R values exceeding 0.85 and IOA values greater than 0.80 (Table S7). These metrics indicate a high level of agreement between observed and modeled data, comparable to results reported in previous studies (Qin et al., 2023; Zheng et al., 2023). The contributions of $R_{NetProd}$ predominantly occur during the daytime (Figure 6). The maximum average daytime $R_{NetProd}$ of OVOCs was observed at ZL ($5.9\pm3.5 \times 10^{-9}$ h$^{-1}$), followed by CQ ($4.11\pm11.9 \times 10^{-9}$ h$^{-1}$), XD ($3.6\pm2.4 \times 10^{-9}$ h$^{-1}$), CD ($3.5\pm2.0 \times 10^{-9}$ h$^{-1}$) and TZ ($1.9\pm3.6 \times 10^{-9}$ h$^{-1}$) sites. This suggests that abundant reactive VOCs emissions at urban areas as well as in the industrial areas could lead to faster generation of OVOCs. Generally, the $R_{NetProd}$ varied with a single peak due to photochemical formation and export transport, with the maximum value at 12:00-14:00 LT. The mean peak of $R_{NetProd}$ at ZL was $8.8 \times 10^{-9}$ h$^{-1}$, followed by XD ($5.6 \times 10^{-9}$ h$^{-1}$), CQ ($5.5 \times 10^{-9}$ h$^{-1}$), CD ($5.1 \times 10^{-9}$ h$^{-1}$) and TZ ($3.0 \times 10^{-9}$ h$^{-1}$). Generally, in the early morning hours (4:00-10:00 LT), positive $R_{Meas}$ at CD, CQ, and XD sites are driven by $R_{Emis\&Tran}$ import. During this period, OVOCs mixing ratios show a significant upward trend, peaking between 8:00 and 10:00 LT.

Overall, OVOCs mixing ratios at CD, CQ, and ZL sites were typically lower at night but higher during the daytime (Figure 6), attributing to strong daytime photochemical generation, especially at around 7:00-10:00 LT. In contrast, TZ and XD showed higher nighttime OVOCs than that at daytime, which is due to stronger emission import during night. In addition, though $R_{NetProd}$ at ZL was the fastest during the daytime, the airmass transport can efficiently export OVOCs to downwind areas, leading to relatively lower OVOCs mixing ratios. While at TZ, $R_{NetProd}$ was the lowest, but the

daytime OVOCs was the highest due to the predominant daytime import, especially the
southwestward import on August 8 (Figure S5 (f)).

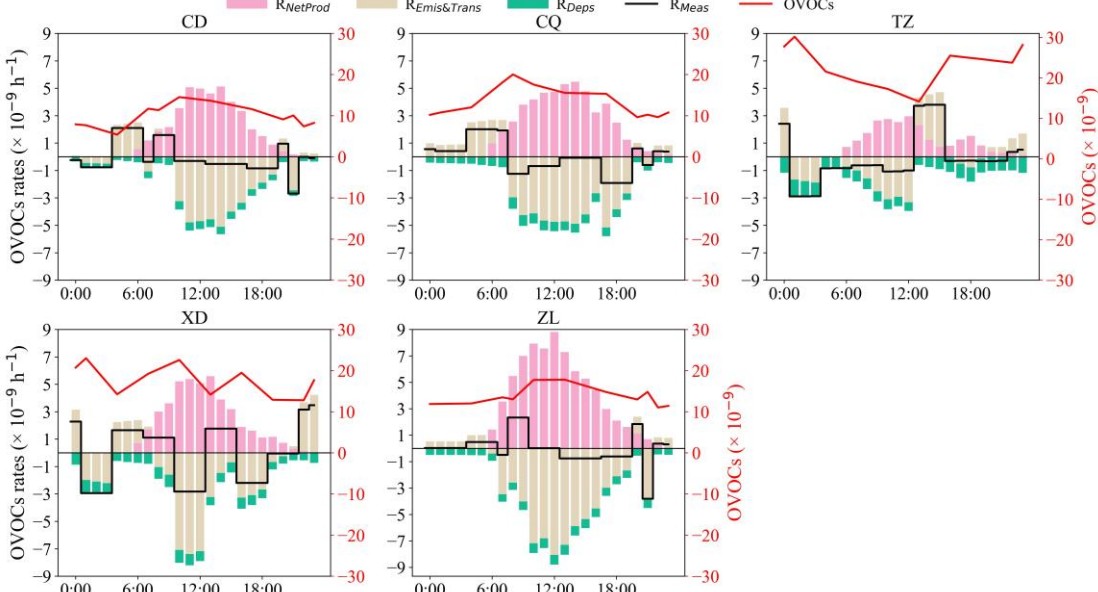

**Figure 6. Diurnal profiles of OVOCs contributions from local photochemical production ($R_{NetProd}$),**
**deposition ($R_{Deps}$), emission/transport ($R_{Emis\&Trans}$), and measured OVOCs formation rate ($R_{Meas}$) of the five**
**sites.**

## 3.3 Importance of OVOCs observationally constraint in OBM

To investigate the effect of the constrain setting of OVOCs on OBM performance, the
simulated OVOC, $O_3$, radicals in the Free and Base scenarios are compared (Figure 7). It has been
shown that the box model, which did not take into account transport (including horizontal and
vertical diffusion) and emissions, will result in overestimations of OVOCs, peroxyl radical and PAN
(Qu et al., 2021). In this study, OVOCs are overestimated by 42.1%~126.5% in the Free scenario
compared with the Base scenario (Figure 7 (a), Figure S6 (c)), especially HCHO (76.3%) and
benzaldehyde (737.5%). The daytime $RO_x$ was overestimated by 6.5%~23.3%, with $RO_2$ and $HO_2$
being overestimated by 6.6%~35.1% and 5.3%~20.4%, respectively, while OH was underestimated
by 1.8%~20.9% (Figure 7 (d-f), Figure S7 (b)). As shown in Figure 8 (a), photolysis of OVOCs
(include HCHO) is the predominant source of $RO_x$ radicals ($P(RO_x)$) in the daytime, which is
consistent with the findings in Beijing (Liu et al., 2012), Shanghai (Zhang et al., 2021a), Hong Kong
(Xue et al., 2016), and Mexico (Volkamer et al., 2010). To assess the impact of OVOCs on the
simulation of $RO_x$ species ($RO_2$, $HO_2$, and OH), the chemical budgets of these species, as influenced
by OVOCs, are quantified according to Liu et al. (2012) and Xue et al. (2016) (Figure 8 (b), Figure
S8 (a)).
In the Free scenario, the daytime net production of $RO_x$ ($P(RO_x)$) was estimated to range from
0.03 to $0.14 \times 10^{-9}$ h$^{-1}$ compared to the Base scenario across four sites (excluding TZ), indicating an
overestimation of $RO_x$. Notably, the TZ site exhibited negative $P(RO_x)$ values, suggesting the
potential existence of unaccounted $RO_x$ sources in this region. The mean daytime $P(RO_x)$ in the Free
scenario was calculated as $4.8\pm2.7 \times 10^{-9}$ h$^{-1}$, 18.8% higher than that in the Base scenario ($4.0\pm2.3$
$\times 10^{-9}$ h$^{-1}$). As illustrated in Figure 8 and Figure S8(a), the photolysis of OVOCs (including HCHO)
dominants $P(RO_x)$, with a mean rate of $2.9\pm1.9 \times 10^{-9}$ h$^{-1}$ in the Free scenario, 27.4% higher than
that in Base scenario ($2.3\pm1.5 \times 10^{-9}$ h$^{-1}$). This substantial increase in OVOCs photolysis
consequently amplified the formation of peroxyl radicals ($RO_2$ and $HO_2$). Among the production
pathways, the photolysis of HCHO demonstrated the most pronounced impact on $HO_2$ production
in the Free scenario ($0.1\sim1.9 \times 10^{-9}$ h$^{-1}$), with an increase of 7.8%~151.2% ($0.1\sim1.2 \times 10^{-9}$ h$^{-1}$) than
in the Base scenario ($0.5\sim1.1 \times 10^{-9}$ h$^{-1}$) (Figure 8 (b)).
The interference of OVOCs on OH is comprehensive. On the one hand, increased OVOCs
tends to elevate the generation of $HO_2$, which can directly or indirectly boost OH generation via the
reaction of NO (Figure S9). On the other hand, the higher OVOCs levels can decrease OH via the
reaction of OH+OVOCs (Qu et al., 2021; Tan et al., 2019b). In the Free scenario, total OH sources
(including $H_2O_2$+hv, HONO+hv, $O_3$+hv, and $HO_2$+NO) is $7.5\sim12.1 \times 10^{-9}$ h$^{-1}$, which is $0.3\sim1.1 \times$
$10^{-9}$ h$^{-1}$ higher than that in the Base scenario (10.2~11.0 × $10^{-9}$ h$^{-1}$) (Figure S9). Conversely, OH
destruction to peroxyl radicals in the Free scenario (7.1~11.8 × $10^{-9}$ h$^{-1}$) is 0.3~2.1 × $10^{-9}$ h$^{-1}$ higher
than that in Base scenario (6.1~9.7 × $10^{-9}$ h$^{-1}$), leading to a net OH loss of 0.1~0.9 × $10^{-9}$ h$^{-1}$. This
underestimation of OH without OVOCs constraint biases atmospheric oxidation capacity (AOC) by
0.1%~10.0% (excluding XD) (Figure S10), affecting the evaluation of VOCs decay via OH
oxidation (Li et al., 2022).

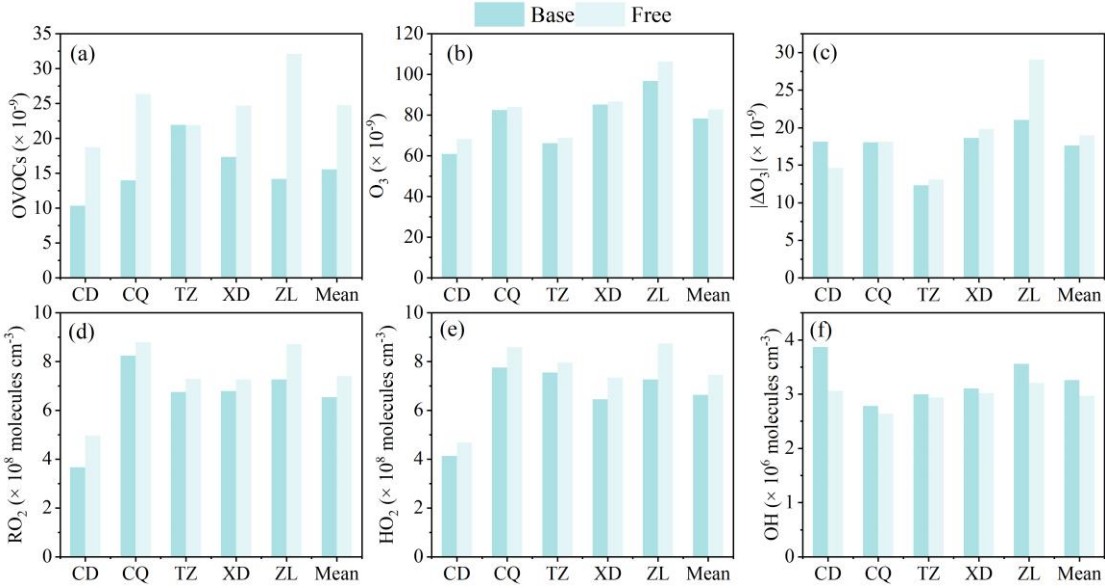


**Figure 7. Comparison of average (a) OVOC, (b) daytime O$_3$, (c) daytime |ΔO$_3$| (the gap between simulated**
**and observed daytime (8:00-18:00 LT) O$_3$ mixing ratios), and RO$_x$ ((d) RO$_2$, (e) HO$_2$, and (f) OH) between**
**the Base and Free scenario simulations.**

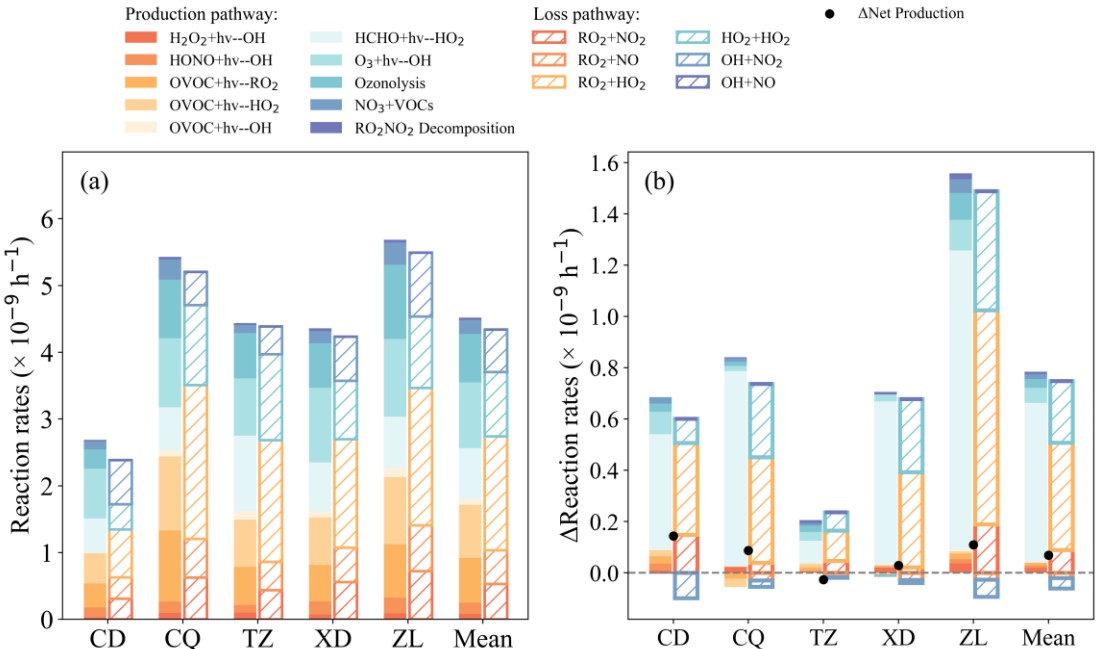

Figure 8. (a) Average daytime (8:00-18:00 LT) sources and sinks of $RO_x$ in Base case and (b) the impact of OVOCs observationally constraints on $RO_x$ budget, calculated by the difference between the Free and Base scenarios.

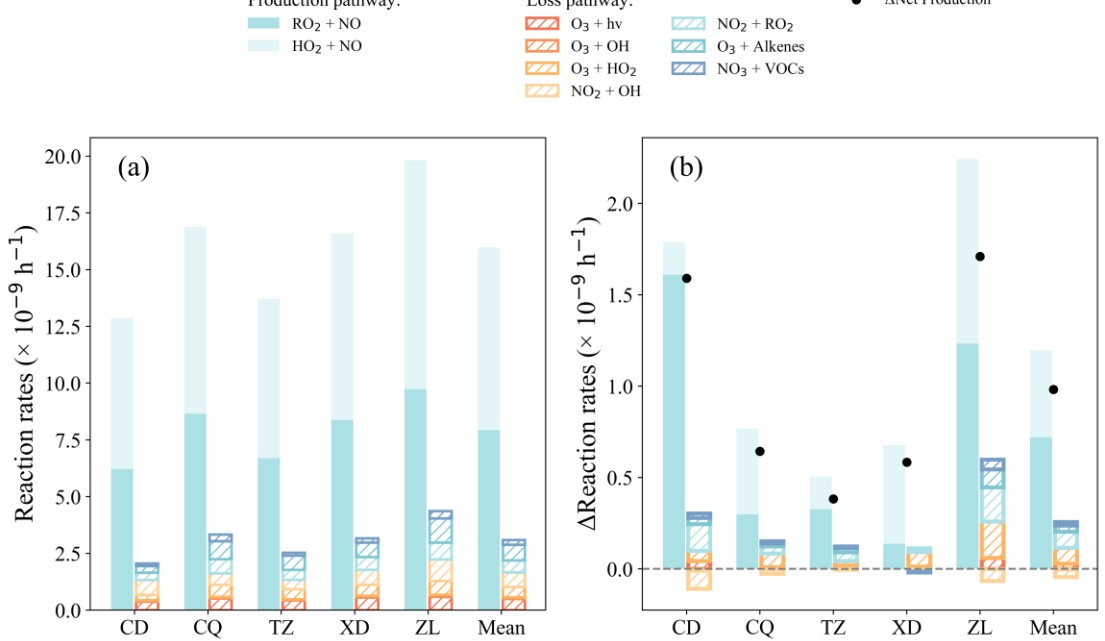

Figure 9. (a) Average daytime (8:00-18:00 LT) $O_3$ production and loss rates ($\times 10^{-9}$ h$^{-1}$) in Base scenario and (b) the impact of OVOCs observationally constraints on $O_3$ budget, calculated by the difference between the Free and Base scenarios.

Overall, the gap between simulated and observed daytime $O_3$ mixing ratios ($|\Delta O_3|$) in the Free case increased by 7.5% compared to that in the Base scenario. To better understand the influence of

OVOCs on $O_3$ formation, the rates of the main production and sink pathways of $O_3$ in the Base and
Free scenarios are summarized (Figure 9, Figure S8 (b)). Compared to the Base scenario, the diurnal
$P(O_3)$ in the Free scenario increases by 3.7%~13.9%, with the reaction rates of $RO_2$+NO and
$HO_2$+NO increases by 1.6%~25.9% ($0.1~1.6 \times 10^{-9}$ h$^{-1}$) and 2.5%~10.0% ($0.2~1.0 \times 10^{-9}$ h$^{-1}$),
respectively. This is attributed to the elevated $RO_2$ and $HO_2$ radical concentrations simulated without
constraints of OVOCs observations (Figure 7 (d-e)). In contrast, the lower simulated concentration
of OH radicals in the Free scenario results in a 0.5%~14.8% ($0~0.1 \times 10^{-9}$ h$^{-1}$) decrease in the
OH+$O_3$/$NO_2$ reaction rate. Although the $L(O_3)$ in the Free scenario is $0.4~1.7 \times 10^{-9}$ h$^{-1}$ higher than
that in the Base scenario, it cannot offset the increase of $P(O_3)$, leading to higher net product of $O_3$.
Thus, no constraints of OVOCs could lead to overestimate peroxyl radicals in the OBM, which in
turn significantly overestimates the deviation of $O_3$ due to the formation pathway of $RO_2$/$HO_2$+NO.
Of course, the impact of OVOCs varies considerably in different emission and functional regions.
The simulated mean daytime $O_3$ in the Free scenario is 106.3, 86.7, 84.1, 68.8, and $68.2 \times 10^{-9}$ h$^{-1}$
at ZL, XD, CQ, TZ, and CD, respectively, which is 9.8%, 1.8%, 2.0%, 3.9%, and 11.9% higher than
that in the Base scenario (Figure 7 (b), Table S8). The mean diurnal $|\Delta O_3|$ at ZL site ($29.1 \times 10^{-9}$,
38.2%) in the Free scenario was significantly higher compared to that of the Base scenario, followed
by XD ($19.8 \times 10^{-9}$, 6.4%), TZ ($13.1 \times 10^{-9}$, 6.2%) and CQ ($18.1 \times 10^{-9}$, 0.6%) (Figure 7 (c), Table
S8), suggesting that the absence of OVOCs constraint in OBM can significantly bias the $O_3$
formation analysis in urban and industrial areas with complex emissions.
**3.4 Uncertainty analysis**
The uncertainty of the model mainly comes from the setting of $NO_x$ (Text S1). Sensitive model
runs are performed with different $NO_x$ settings to show the corresponding uncertainty. Firstly, $NO_{ss}$,

VOCs, OVOCs, T, RH, and BLH are constrained by the observed data, and sensitivity simulations

were carried out for gradient $NO_2$ mixing ratios (50%, 60%, 70%, and 100% of observed $NO_2$

mixing ratios). At the four sites (except TZ), different $NO_2$ settings produce uncertainties of -

12%~25% for daytime $O_3$ and -17%~36% for the daytime $R_{NetProd}$ of OVOCs. In addition, different

$NO_2$ settings produce uncertainties of -27%~51% for the $R_{Emis\&Trans}$ of OVOCs. This indicates that

reducing the uncertainty of $NO_2$ observations is important for further atmospheric chemistry

modelling. Secondly, the $NO_{ss}$ and observed NO mixing ratio was respectively used in the model to

investigate the influence of nearby NO emissions on the OBM result (Table S3). Sensitivity tests

show that directly using the observed NO data may resulted in an uncertainty of 3% ~20%, 10%

~27% and 7.1%~38% for $O_3$, $R_{NetProd}$ and $R_{Emis\&Trans}$, respectively, with significantly higher

uncertainties at TZ than at other stations, which may be related to the strong transient emissions of

$NO_x$.

The OH radicals are the main oxidant in the atmospheric troposphere. Most trace gases (CO,

$CH_4$ and VOCs) react with OH to produce peroxyl radicals, including $HO_2$, and $RO_2$. Peroxyl

radicals then react with NO to promote $O_3$. Photolysis of HONO is one of the important sources of

OH radicals. However, direct measurement of HONO was not conducted in this study. The input of

HONO in this study used the value of linear relationship with $NO_2$ mixing ratios, which may cause

some uncertainty. Sensitivity analyses were conducted based on the typical urban site (ZL) to

quantify the effect of different $HONO/NO_2$ ratios on free radicals and $O_3$. As shown in Figure S12,

as the ratio of $HONO/NO_2$ increased, the mixing ratio of HONO increased, and the concentration

of OH produced by photolysis of HONO increased. At the same time, the peroxyl radical and $O_3$

concentrations also increased (Table S9). At the lowest $HONO/NO_2$ ratio (0.005), the daytime OH,

HO$_2$, RO$_2$, and O$_3$ levels decreased by 4.6%, 4.8%, 6.0%, and 3.4%, respectively, compared to the
HONO/NO$_2$ ratio of 0.02. On the contrary, at the highest HONO/NO$_2$ of 0.04, the daytime OH, HO$_2$,
RO$_2$, and O$_3$ levels increased by 5.4%, 5.6%, 7.2% and 3.8%, respectively, compared to the Base
case.
When analyzing the contribution of chemical pathways, emission/transport and deposition to
OVOCs according to Equations (1-3), the OVOC budget may be affected by the modeling time step.
A sensitivity analysis was conducted at ZL, with simulations at different time steps (30 min, 10 min,
and 5 min). As shown in Figure S13, the diurnal trends of the chemical contributions (R$_{NetProd}$),
emission/transport (R$_{Emis\&Trans}$) and deposition (R$_{Deps}$) to OVOC are similar. The magnitude of the
instantaneous change in R$_{Emis\&Trans}$ decreases when the time step is shortened. Specifically, the
contributions of R$_{NetProd}$ and R$_{Emis\&Trans}$ to OVOC increased, while the contribution of R$_{Deps}$
decreased with shorter time steps (Table S10). When the time step was reduced to 5 minutes, the
contributions of R$_{NetProd}$ and R$_{Emis\&Trans}$ to OVOC increased by 4.3% and 5.0%, respectively, while
the contribution of R$_{Deps}$ decreased by 0.2%. Therefore, shortening the time step in the model
simulation may result in limited increase contribution from R$_{NetProd}$ and R$_{Emis\&Trans}$ to OVOC.
The OBM, while advantageous for its use of the near-explicit master chemical mechanism and
its ability to reproduce atmospheric chemistry based on observations, has several inherent
limitations. Firstly, it inadequately considers emission and transport processes (Wolfe et al., 2016),
leading to uncertainties and potential biases in atmospheric chemical analysis. Future studies could
differentiate the two processes using CTMs. Secondly, the steady-state assumption, which underpins
OBM calculations, might become invalid in areas strongly influenced by nearby emissions (e.g.,
high-traffic and industrial intensive regions) (Wolfe et al., 2016). This may potentially cause
overestimations of secondary products such as $O_3$, $RO_x$ radicals, and OVOCs (Li et al., 2021, 2014b).
In this study, simulations use $NO_{ss}$ as inputs, acknowledging that traffic-related NO emissions may
prevent the system from reaching an approximate steady state. The associated uncertainties have
been discussed above. Lastly, the model may not fully capture all relevant chemical processes, such
as heterogeneous reactions or the formation and the fate of intermediate species. For instance, the
exclusion of $ClNO_2$ chemistry could result in underestimating $RO_x$ and $O_3$ production in certain
environments (Riedel et al., 2014; Xia et al., 2020). Future research should focus on refining the
chemical mechanisms to better approximate real atmospheric conditions.

## 4. Conclusions

Compared with previous studies conducted in most Chinese cities, the VOCs level in Zibo is
in the upper-middle range ($>32 \times 10^{-9}$), with OVOCs being the second-largest contributor
(29.4%~36.1%) after alkanes (34.8%~53.3%). Higher levels of OVOCs were observed at sites with
more prominent emissions, with mixing ratios ranked as suburban (TZ, $19.7 \times 10^{-9}$) > industrial
(XD, $16.8 \times 10^{-9}$) > urban (ZL, $14.9 \times 10^{-9}$) > upwind (CQ, $13.9 \times 10^{-9}$) > downwind (CD, $10.4 \times$
$10^{-9}$). The OFP in Zibo is $410.4\pm197.2$ µg m$^{-3}$, with OVOCs accounting for the largest proportion
(31.5%~55.9%). The upwind site (CQ, $464.2\pm162.3$ µg m$^{-3}$) has the highest OFP, followed by the
suburban site (TZ, $456.3\pm295.3$ µg m$^{-3}$), the urban site (ZL, $441.1\pm174.5$ µg m$^{-3}$), the industrial site
(XD, $422.9\pm166.9$ µg m$^{-3}$), and the downwind site (CD, $279.4\pm101.2$ µg m$^{-3}$) (Table S5). OFP
contributed by OVOCs is most dominant at suburban (TZ, 254.9 µg m$^{-3}$) and industrial (XD, 194.7
µg m$^{-3}$) sites, followed by urban (ZL, 148.9 µg m$^{-3}$), upwind (CQ, 146.2 µg m$^{-3}$), and downwind
(CD, 102.3 µg m$^{-3}$) sites. The high levels of OVOCs and their significant contribution to OFP
highlights their crucial role in $O_3$ production across the observing stations.

Based on the OBM simulation, daytime OVOCs primarily originate from photochemical generation, while at nighttime, emissions/transport is the main sources. This diurnal pattern is closely related to the cyclical nature of human activities in urban areas (ZL), where stronger human activities such as vehicle emissions in the daytime enhance the secondary generation of OVOCs. Conversely, in industrial and suburban areas (XD, CQ, and TZ), emissions/transports dominate nighttime OVOC levels, leading to higher mixing ratios at night compared to the daytime.

Without OVOCs constraint in the OBM, OVOCs are overestimated by 42.1%~126.5% in the Free scenario. The impact of OVOCs constraint on $P(RO_x)$ is most significant at the urban site (ZL) (29.4%), comparable to downwind site (CD) (27.6%), and higher than the industrial site (XD) (17.8%), upwind site (CQ) (15.8%) and suburban site (TZ) (4.7%). In addition, this overestimation of OVOCs in the Free scenario accelerates the reaction of OH with OVOCs and the photolysis of OVOCs, promoting increased production of $RO_2$ (6.6%~35.1%) and $HO_2$ (5.3%~ 20.4%), which in turn leads to an overestimation of $O_3$ (1.8%~11.9%) during the daytime. However, the reaction rates of OH with OVOCs are overestimated without OVOCs constraint, which leads to underestimation of OH (3.4%~12.7%) and AOC (0.1%~10.0%). Therefore, to minimize the bias of numerical models, particularly in areas with complex anthropogenic activities, it is essential to intensify OVOCs observations and integrate them into numerical models. These efforts are crucial for refining atmospheric photochemistry simulation, improving the accuracy of $O_3$ formation predictions, and formulating more effective air quality management strategies for regions experiencing similar pollution challenges.

## Author contributions

KZ, XY, RL, JX, QL, LSS, JQL, YNY, FTW and LMY conducted the field measurements.

JWD and KZ performed the data analysis and prepared the manuscript with contributions from all
co-authors. KZ, YLF, HC, LH, JNT, YJW and LL reviewed and edited the manuscript. All authors
contributed to data interpretation and discussions.

## Data availability

The model output is available upon request by contacting the corresponding author.

## Author contributions

LL conceptualized this study along with ZK. YX, LR, XJ, LQ, SLS, LJQ, YYN, WFT and YLM did
the field observation. Data analysis and model simulation were done by ZK and DJW. The
manuscript draft was prepared by DJW, and editing was done by ZK, LL, FYL, CH, HL, TJN and
WYJ.

## Competing interests

The contact author has declared that none of the authors has any competing interests.

## Acknowledgment

This study was financially supported by the National research program for key issues in air
pollution control (DQGG202119). This work is supported by Shanghai Technical Service Center of
Science and Engineering Computing, Shanghai University.

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
