# Peer review of "on atmospheric chemistry: A case study in a typical industrial city in"

_EGUsphere, 2024_

## Author Comment (AC1)

**Response to Reviewers comments:**

**Reviewer #2**: The article from Dai et al., "Significant influence of oxygenated volatile organic compounds on atmospheric chemistry analysis: A case study in a typical industrial city in China", investigates the contribution of volatile organic compounds, including oxygenated VOC, to ozone formation potential using ambient data from five observation sites around Zibo, an industrial city in the North China Plain, and an observation-based model (OBM).

The topic covered by the study can help assessing correct strategies to mitigate urban air pollution, therefore I find the study of interest to the atmospheric sciences community and in scope with the journal. However, in its current form, the methodology used by the authors is not described with enough clarity and for this reason it is hard to assess the robustness of the study and the presented results. For this reason, I would suggest publication after some major revisions are considered.

Response: We thank the reviewer for the positive and constructive comments. Below is our point-by-point response to each comment, marked in blue. Changes made to the main text are marked in blue in the revised manuscript file.

**General comments:**

1. The main comment I have concerns the methodology used by the authors. The experimental methods used by the authors are not sufficiently presented. The authors use measurements from five observation sites collected during five days using online gas-chromatography (GC) and offline gas-chromatography. While offline GC can require extensive timing for both sampling and analysis and can be limited by samples availability, I wonder if there is any particular reason why the online field campaigns lasted for short time periods and if the limited sampling time is representative to the field

sites and conditions? Can the authors explain this choice, maybe related to previous studies conducted in the same area? The authors should also provide some information about the sampling and analytical methods used (type of analysis, GC column characteristics, desorption and separation methods, sampling inlets characteristics, use of any ozone scrubber during the sampling, samples storage) and uncertainty associated to the results. As the whole discussion is based on comparing the results among different sites there should be mention that the analytical techniques and sampling are comparable, was any test conducted to prove that?

Response: Thank you for the detailed feedback and for highlighting the importance of clarifying the methodology. We understand the concerns regarding the representativeness and comparability of our sampling and analytical methods.

For question 1, The short-term sampling campaign conducted in August 2021 was strategically designed to capture VOC and OVOC concentrations across multiple sites during a representative ozone pollution episode. To ensure comprehensive temporal coverage, we collected 1-3 hourly samples at each station, resulting in 8-10 samples per station every day, which effectively captured the diurnal variability of atmospheric composition. Throughout the observation period, we obtained a total of 39 valid samples from both the suburban (TZ) and industrial (XD) stations, while the upwind (CQ), downwind (CD), and urban (ZL) stations yielded 49 valid samples each. This sampling strategy provided robust datasets that accurately represent the spatial and temporal variations of VOC and OVOC concentrations during the study period.

For question 2, we have revised the descriptions of VOC and OVOC measurements methods. Please refer to Lines 129-151:

"A mixture of 56 PAMS target species (Spectra Gases Inc., USA, Table S2) was used for the

calibration of the GC-FID system. Each VOC analyzer provided measurements with a 1-hour temporal resolution. More detailed descriptions of these instruments can be found in previous studies (Li et al., 2023; Wang et al., 2014; Yang et al., 2022; Zheng et al., 2023). Overall, the detection limits for most VOC species are below 0.1 ppb. Additionally, 18 oxygenated VOCs species were collected by 2,4-dinitrophenylhydrazine (DNPH) sorbent tubes in conjunction with an automated sampler for a period of 1 or 3 hour per sample. A 47 mm quartz filter membrane is attached to the front of the sampling tube to filter particulate matter. An ozone scrubber (silica gel column tubing coated with potassium iodide) was placed at the front of the air inlet to avoid ozone interference. OVOCs were derivatized in cartridges to hydrazones during sampling. The cartridges were eluted with 3 ml of acetonitrile and stored at 0-4 °C immediately. Then the eluants were analyzed using an Agilent HPLC, equipped with ultraviolet absorption detector (UVD), quadruple pump, and Agilent TM C18 reversed column (250 mm×4.6 mm, 5.0 μm). A gradient elution was used, and the mobile phase was mixing of acetonitrile, tetrahydrofuran and water. The analysis was carried out using a ternary gradient elution program at a flow rate of 1.2 mL/min, with detection wavelength of 360 nm, and sample volume of 10 μL at a column temperature of 45 °C. More details about OVOC samplings and analysis can be found in Peng et al. (2023). The lower limit of detection for OVOCs were <0.1 ppb (Peng et al., 2023). A total of 271 valid OVOCs samples were collected during the campaign. At the industrial (XD) and suburban (TZ) stations, 8 samples were collected per day at 3-hour intervals. At the urban (ZL), upwind (CQ) and downwind (CD) stations, 10 samples were collected per day, with 7 samples collected at 1-hour intervals during 7:00-21:00 LT, and 3 samples collected at 3-hour intervals during the night (1:00-6:00 and 22:00-1:00$^{+1}$ LT), and totaling 59 valid samples per station."

For question 3, while we did not conduct specific tests to evaluate inter-site comparability. However, the VOC analyzing instruments used at the five sites were operated and maintained by the same organization, following standardized protocols. These instruments were regularly calibrated using identical standard gas mixtures to maintain measurement accuracy and consistency. For offline OVOC sampling, we employed uniform methodologies across all sites, including the use of identical sampling instruments, standardized collection techniques, consistent sample preservation procedures, and the same analytical protocols. This comprehensive approach to quality control and methodological standardization ensures that the observational data from the five sites are directly comparable and maintain consistent data quality throughout the study period.

2.  The calculation-modelling methods miss some information as well. It is not easy to follow section 2.2 as many acronyms are present and several assumptions considered. Could the authors adjust this section in a way that it is clear which assumptions are considered and why? Could the authors provide the different equations used in their calculations and model with more detail? There is no mention on how the ozone formation potential is calculated and which data are used for that.

Response: Thank you for your detailed feedback regarding the calculation and modeling methods in Section 2.2 in the origin manuscript. We apologize for any confusion or questions that may have arisen due to the unclean expression. We recognize the importance of providing clear and comprehensive explanations for readers to fully understand our methodology.

(1) Clarification of assumptions and calculation methods of OVOC budget:

In our analysis of the OVOC budget, we assume that variations in the mixing ratios of OVOCs

within a given air mass are driven by four factors: chemical production and loss, emissions, transport, and deposition. This assumption underpins our calculations and is discussed in added Section 2.3, Lines 201-218, where the relevant descriptions and methodologies are introduced in detail:

"At a given site, variations in OVOCs mixing ratios are mainly influenced by in-situ photochemical production and chemical loss, emissions, regional transport, and deposition (Tan et al., 2018a; Xue et al., 2014a; Zhang et al., 2021). The change rate of observed OVOCs ($R_{Meas}$) is calculated by Equation (1). The in-situ photochemical production of OVOC ($R_{ChemProd}$) is mainly caused by the oxidation of VOCs, while their in-situ chemical loss ($R_{ChemLoss}$) includes photolysis and reactions with oxidants (OH, $NO_3$, and $O_3$) (https://mcm.york.ac.uk/MCM/, last access: 13 Jan 2025) (Atkinson, 2000; Atkinson and Arey, 2003; Jenkin et al., 2015; Saunders et al., 2003). The in-situ net OVOCs chemical production ($R_{NetProd}$) (Equation (2)) and their removal by deposition ($R_{Dep}$) are calculated hourly according to the OBM simulation. The OBM primarily accounts for atmospheric photochemical reactions, and deposition within the boundary layer. However, previous studies have reported that the OBM lacks an explicit representation of transport processes and emissions (Wolfe et al., 2016; Zhang et al., 2021), making it challenging to disentangle their respective contributions. Therefore, emissions and transport are combined to a single term ($R_{Emis\&Trans}$) to represent their contributions collectively. If the $R_{Emis\&Trans}$ is positive, it is considered a net import of emissions/transport, whereas a negative suggests a net export. The emissions and regional transport of OVOCs ($R_{Emis\&Trans}$) are computed as Equation (3).

$$R_{Meas} = \sum_i \frac{d([OVOC]_i)}{dt} \tag{1}$$

$$R_{NetProd} = \sum_i (R_{ChemProd,i} - R_{ChemLoss,i}) \tag{2}$$

$$R_{Emis\&Trans} = (R_{Meas} - R_{NetProd} - \sum_i R_{Dep,i}) \tag{3}$$

where [OVOC]$_i$ is the mixing ratios of OVOC species i constrained in OBM, 15 in total (Table S2).

dt is the time-step of modeling, d[OVOC]$_i$ refer to the change in mixing ratio of OVOC species i."

(2) Description of the methodology for calculating ozone formation potential (OFP):

To calculate the potential of VOCs contributed to O$_3$ formation, we used the Maximum Incremental Reactivity (MIR) approach. The relevant methodology description has been added. Please refer to Section 2.4, Lines 233-240:

"**2.4 Evaluation of ozone formation potential and atmospheric oxidation capacity**

Different VOC species vary in their capability to form ozone, and their potential to produce O$_3$ can be evaluated by the maximum incremental reactivity (MIR) (Carter, 2010). The ozone formation potential (OFP) calculated for each VOC species represents its maximum contribution to ozone production (Bufalini and Dodge, 1983). The OFP of VOCs is calculated as follows:

$$OFP_i = [VOC]_i \times MIR_i \tag{6}$$

where OFP$_i$ is the OFP of VOC species $i$ ($\mu$g m$^{-3}$), [VOC]$_i$ is the atmospheric concentration of VOC species $i$ ($\mu$g m$^{-3}$), MIR$_i$ is the maximum incremental reactivity coefficient of the VOC species $i$ (g O$_3$/g VOCs) (Table S2) from Carter, 2010a."

The MIR data of VOC species has been added, please refer to Table S2 in the supplement information:

**Table S1 VOCs species and their names in Master Chemical Mechanism (MCMv3.3.1), minimum detection limits (MDL), and maximum incremental reactivity coefficient (MIR).**

**"—" means that the species is not listed in the mechanism.**

| Species | MCM name | MDL (ppb) | MIR | Species | MCM name | MDL (ppb) | MIR |
|---|---|---|---|---|---|---|---|
| **Alkanes** | | | | **BVOCs** | | | |
| Ethane | C2H6 | 0.079 | 0.28 | Isoprene | C5H8 | 0.02 | 10.61 |
| Propane | C3H8 | 0.046 | 0.49 | **Alkynes** | | | |
| Isobutane | IC4H10 | 0.022 | 1.23 | Acetylene | C2H2 | 0.032 | 0.95 |
| n-Butane | NC4H10 | 0.027 | 1.15 | **Aromatics** | | | |
| Cyclopentane | — | 0.016 | 1.15 | Benzene | BENZENE | 0.012 | 0.72 |
| Isopentane | IC5H12 | 0.087 | 2.39 | Toluene | TOLUENE | 0.013 | 4.00 |
| n-Pentane | NC5H12 | 0.031 | 1.31 | Ethylbenzene | EBENZ | 0.014 | 3.04 |
| 2,2-Dimethylbutae | M22C4 | 0.014 | 1.17 | m-Xylene | MXYL | 0.027 | 9.75 |
| 2,3-Dimethylbutane | M23C4 | 0.019 | 0.97 | Styrene | STYRENE | 0.014 | 1.73 |
| 2-Methylpentane | M2PE | 0.031 | 1.50 | o-Xylene | OXYL | 0.012 | 7.64 |
| 3-Methylpentane | M3PE | 0.012 | 1.80 | Isopropylbenzene | IPBENZ | 0.014 | 2.52 |
| n-Hexane | NC6H14 | 0.011 | 1.24 | n-Propylbenzene | PBENZ | 0.013 | 2.03 |
| Methylcyclopentane | — | 0.011 | 2.19 | m-Ethyltoluene | METHTOL | 0.032 | 7.39 |
| 2,4-Dimethylpentane | — | 0.013 | 1.55 | p-Ethyltoluene | PETHTOL | 0.014 | 4.44 |
| Cyclohexane | CHEX | 0.016 | 1.25 | 1,3,5-Trimethylbenzene | TM135B | 0.012 | 11.76 |
| 2-Methylhexane | M2HEX | 0.012 | 1.19 | 1,2,4-Trimethylbenzene | TM124B | 0.011 | 8.87 |
| 3-Methylhexane | M3HEX | 0.013 | 1.61 | 1,2,3-Trimethylbenzene | TM123B | 0.011 | 11.97 |
| 2,3-Dimethylpentane | — | 0.013 | 1.34 | o-Ethyltoluene | OETHTOL | 0.013 | 5.59 |
| 2,2,4-Trimethylpentane | — | 0.012 | 1.26 | m-Diethylbenzene | — | 0.011 | 7.10 |
| n-Heptane | NC7H16 | 0.012 | 1.07 | p-Diethylbenzene | — | 0.011 | 4.43 |

| Species | MCM name | MDL (ppb) | MIR | Species | MCM name | MDL (ppb) | MIR |
|---|---|---|---|---|---|---|---|
| Methylcyclohexane | — | 0.011 | 1.70 | **OVOCs** | | | |
| 2,3,4-Trimethylpentane | — | 0.013 | 1.03 | Formaldehyde | HCHO | 0.007 | 9.46 |
| 2-Methylheptane | — | 0.013 | 1.07 | Acetaldehyde | CH3CHO | 0.016 | 6.54 |
| 3-Methylheptane | — | 0.013 | 1.24 | Acetone | CH3COCH3 | 0.009 | 0.36 |
| n-Octane | NC8H18 | 0.012 | 0.90 | Acrolein | ACR | 0.008 | 7.45 |
| n-Nonane | NC9H20 | 0.013 | 0.78 | Propionaldehyde | C2H5CHO | 0.026 | 7.08 |
| n-Decane | NC10H22 | 0.011 | 0.68 | Crotonaldehyde | C4ALDB | 0.042 | 9.39 |
| n-Undecane | NC11H24 | 0.018 | 0.61 | Butyraldehyde | C3H7CHO | 0.048 | 5.97 |
| n-Dodecane | NC12H26 | 0.048 | 0.55 | Benzaldehyde | BENZAL | 0.055 | -0.67 |
| **Alkenes** | | | | Cyclohexanone | CYHEXONE | 0.058 | 1.35 |
| Ethylene | C2H4 | 0.057 | 9.00 | 3-Methylbutyraldehyde | C3ME3CHO | 0.058 | 4.97 |
| Propylene | C3H6 | 0.022 | 11.66 | Pentanal | C4H9CHO | 0.038 | 5.08 |
| trans-2-Butene | TBUT2ENE | 0.013 | 15.16 | o-Tolualdehyde | OXYLAL | 0.072 | 0.00 |
| 1-Butene | BUT1ENE | 0.023 | 9.73 | m-Tolualdehyde | MXYLAL | 0.089 | 0.00 |
| cis-2-Butene | CBUT2ENE | 0.016 | 14.24 | Hexaldehyde | C5H11CHO | 0.060 | 4.35 |
| trans-2-Pentene | TPENT2ENE | 0.012 | 10.56 | Heptaldehyde | C6H13CHO | 0.034 | 3.69 |
| 1-Pentene | PENT1ENE | 0.093 | 7.21 | Octanal | — | 0.029 | 3.16 |
| cis-2-Pentene | CPENT2ENE | 0.011 | 10.38 | Nonanal | — | 0.032 | 0.00 |
| 1-Hexene | HEX1ENE | 0.014 | 5.49 | Decanal | — | 0.035 | 0.00 |

3. I suggest also to improve harmonization between the main text's manuscript and the supplementary information. In the current form, there are many points discussed in the main text that are not immediately clear to the reader, as the main text misses to recall the corresponding information or figure actually presented in the supplement. Some of the information presented in the supplement could be included in the main text to improve clarity.

Response: Thank you for your valuable suggestion. We have reorganized the key information to present the key charts in the main text's manuscript. The adjusted charts have been marked in blue in the main text's manuscript.

**Specific comments:**

1. line 16. What type of stations?

Response: To clarify, the five stations mentioned in Line 16 are urban, suburban, industrial, upwind, and downwind stations. The relevant descriptions have been revised, please refer to Lines 16-18:

"In this study, 74 VOCs (including 18 OVOCs) were measured at five representative stations (urban, suburban, industrial, upwind, and downwind stations) in Zibo, an industrial city in the North China Plain."

2. Line 18. Previous studies conducted where?

Response: Previous studies conducted most in Chinese cities, offering a statistical basis for comparison.

The related description has been revised, please refer to Lines 18-19:

"The VOCs level in Zibo (44.6±20.9 ppb) is in the upper-middle range (> 32 ppb) compared to previous studies conducted in most Chines cities, with OVOCs contributing for 30.0%~37.8%."

3. Line 81. 5-day field campaign: are they representative to the sites and conditions?

Response: Yes, we consider these 5 days of observations to be representative.

Firstly, the short-term observations are typical of ozone-polluted days and are representative for analyzing photochemistry. And we collected 1-3 hourly samples at each station, resulting in 8-10 samples per station every day, which effectively captured the diurnal variability of atmospheric composition. Throughout the observation period, we obtained a total of 39 valid samples from both the suburban (TZ) and industrial (XD) stations, while the upwind (CQ), downwind (CD), and urban (ZL) stations yielded 49 valid samples each. This sampling strategy provided robust datasets that accurately represent the spatial and temporal variations of OVOC concentrations during the study period.

Secondly, the VOC analyzing instruments used at the five sites were operated and maintained by the same organization, following standardized protocols. These instruments were regularly calibrated using identical standard gas mixtures to maintain measurement accuracy and consistency. For offline OVOC sampling, we employed uniform methodologies across all sites, including the use of identical sampling instruments, standardized collection techniques, consistent sample preservation procedures, and the same analytical protocols. This comprehensive approach to quality control and methodological standardization ensures that the observational data from the five sites are directly comparable and maintain consistent data quality throughout the study period.

4. Section 2.1 please provide information about the methods as suggested in my general comment

Response: Thanks for your suggestion. We have added information about the methods in Section 2.1, please refer to the General comments 1.

5. Line 107. PAMS=?

Response: PAMS is the Photochemical Assessment Monitoring Stations, including 28 alkanes, 16 aromatics, 10 alkenes, acetylene, and isoprene. The relevant descriptions have been added clearly, please refer to Lines 107-109:

"Site-scale wind patterns can affect the levels and spatial distribution of OVOCs and PAMS (target VOC species from the Photochemical Assessment Monitoring Stations, including include 29 alkanes, 16 aromatics, 9 alkenes, isoprene, and acetylene) across sites."

6. I would suggest to include here a list of the measured compound, and the classification method considered by the authors to recall the chemical compounds in subsequent analysis.

Response: Thank you for your suggestion. We have provided a categorized list of VOCs here. Pease refer to Table S2, please refer to general comments 2.

7. Section 2.2 Please provide information about the methods and calculations as suggested in my general comment

Response: Thanks for your suggestion. We have made the relevant revision under your general comment 2 (1). And the related information about methods and calculations has been moved to Section 2.3, Lines 201-218.

8.  Figure. 2 It is hard to see the contribution from acetylene in the figure.

Response: Thank you for your careful check-up. We have modified the color of acetylene, please refer to Figure 2.

[Figure]

Figure 1. Time profiles of pollutant mixing ratios and meteorological parameters in Zibo from August 8 to 12, 2021. The meteorological data were from ZL, the central site of Zibo, and the pollutants data were the average of the five sites. The hourly PAMS (including alkanes, alkenes, aromatics, acetylene, and isoprene) data were aligned with the 1/3-hour sampling intervals of the OVOCs data to ensure comparability between the two datasets.

9.  Line 195. Is this also seen by the largest concentration of aromatics?

Response: Are you referring to the TZ station? While it is close to the oil field, its aromatic

levels are not the highest among the five stations. However, it does exhibit the highest levels of alkanes.

10. Line 198. Transport from where to where. Is this also the site where the largest concentration of $O_3$ is formed?

Response: For question 1, $O_3$ and its precursors were transported from the northeast to the southwest, i.e., sequentially from upwind (CQ) to urban stations (ZL), and then to downwind stations (CD).

For question 2, yes, $O_3$ concentration at the downwind site (CD) is higher than at any other site. And the $O_3$ concentration increases sequentially from the northeast to southwest (CQ $\rightarrow$ ZL $\rightarrow$ CD).

The relevant descriptions have been revised, please refer to Lines 288-294:

"The downwind site (CD) has slightly lower $NO_2$ level (10.8±5.1 ppb) and lower TVOCs mixing ratios (35.7±12.5 ppb) than urban site (ZL, 14.8±6.5 and 40.6±10.3 ppb, respectively) and upwind site (CQ, 12.7±8.1 and 42.3±15.4 ppb, respectively), and has higher $O_3$ mixing ratio (58.6±30.0 ppb) than CQ and ZL station. This may be attributed to the sequential transport of $O_3$ and its precursors from the upwind station (CQ) to urban station (ZL), and subsequently to the downwind station (CQ), driven by the dominant northeasterly winds (**Error! Reference source not found.** (b), Figure S1)."

11. Line 214. Please provide the method to determine OFP

Response: The method of calculating OFP has been added in Section 2.4, Lines 233-240. Please

refer to general comments 2 (2) for details.

12. Line 215. What are the contributions of other classes of VOC?

Response: In addition to OVOC, aromatics contribute the second largest share of OFP with 10.2%~41.2%. Alkanes (10.3%–24.6%) and alkenes (11.4%~23.1%) contributed comparable proportions, and BVOC (2.1%~7.6%) has the lowest share.

The relevant expressions have been refined, please refer to Lines 308-312:

"The mean OFP in Zibo during the observation is 410.4±197.2 µg m$^{-3}$, with OVOCs accounting for the largest proportion (31.5%~55.9%), followed by aromatics (10.2%~41.2%). Alkanes (10.3%–24.6%) and alkenes (11.4%~23.1%) make comparable proportions, while BVOCs accounted for only 2.1%~7.6% of the total OFP (Figure 5 (b))."

13. Line 216. Which OVOC contributes the largest? The authors can provide here the OVOC speciation.

Response: Thanks for your suggestion. We have added relevant content that complements the top three contributing OVOCs species. Please refer to Lines 328-332:

"Among OVOC species, HCHO is the dominant contributor to OFP across the five sites (56.6~202.0 µg m$^{-3}$). This is consistent with previous studies (Duan et al., 2008; Huang et al., 2020; Zhou et al., 2024). The top four OVOC species are formaldehyde, acetaldehyde, propionaldehyde, and butyraldehyde, which cumulatively contributed 91%~95% of the OFP from OVOCs (Table S5)."

14. Line 233. Please include a table with measured concentrations, OFP from each measured VOC

and VOC group.

Response: We have added a table with measured concentrations, OFP from each measured

VOC and VOC groups. Please refer to Table S5.

**Table S5. Measured mixing ratios, ozone formation potential from VOC species and groups.**

| Species/Groups | VOC mixing ratios (mean±std) (ppb) | | | | | OFP (mean±std) ($\mu g\ m^{-3}$) | | | | |
|---|---|---|---|---|---|---|---|---|---|---|
| | CD | CQ | TZ | XD | ZL | CD | CQ | TZ | XD | ZL |
| TVOCs | 35.7±12.5 | 42.3±15.4 | 58.5±35.0 | 49.6±19.0 | 40.6±10.3 | 279.4±101.2 | 464.2±162.3 | 456.3±295.3 | 422.9±166.9 | 441.1±174.5 |
| Alkanes | 13.2±6.2 | 16.5±8.5 | 30.2±21.0 | 23.3±11.2 | 13.5±5.6 | 36.0±19.3 | 47.8±25.4 | 66.5±39.9 | 103.9±51.7 | 46.7±21.9 |
| Alkenes | 3.3±1.8 | 3.3±1.6 | 2.9±1.7 | 2.8±1.3 | 5.6±3.0 | 57.1±30.4 | 57.5±30.5 | 52.1±30.8 | 50.6±27.2 | 101.8±56.8 |
| Aromatics | 4.0±1.7 | 7.0±3.6 | 2.2±1.2 | 3.1±1.5 | 6.3±4.7 | 73.4±29.1 | 191.3±111.3 | 46.6±24.8 | 48.4±23.0 | 125.3±116.2 |
| OVOCs | 10.7±5.0 | 14.5±6.7 | 22.1±22.5 | 17.9±8.5 | 14.6±4.8 | 102.3±51.2 | 146.2±70.7 | 254.9±276.1 | 194.7±101.0 | 148.9±55.7 |
| Acetylene | 4.4±4.1 | 0.4±0.7 | 0.0±0.0 | 1.9±1.6 | 0.0±0.0 | 4.8±4.5 | 0.5±0.8 | 0.0±0.1 | 2.1±1.7 | 0.0±0.1 |
| Isoprene | 0.2±0.2 | 0.6±0.6 | 1.1±0.8 | 0.7±0.5 | 0.6±0.7 | 5.8±7.7 | 20.8±17.8 | 34.9±25.8 | 23.0±16.3 | 18.1±23.6 |
| Benzene | 1.0±0.5 | 0.3±0.2 | 0.2±0.3 | 1.1±0.6 | 1.2±0.5 | 2.4±1.3 | 0.6±0.4 | 0.4±0.7 | 2.7±1.5 | 3.0±1.3 |
| Toluene | 1.2±0.5 | 1.2±0.9 | 1.4±1.0 | 0.9±0.5 | 1.6±1.3 | 19.3±7.5 | 20.0±15.2 | 23.6±15.9 | 14.5±8.2 | 25.8±21.4 |
| Ethylbenzene | 0.3±0.2 | 0.4±0.5 | 0.1±0.1 | 0.2±0.1 | 0.7±0.3 | 4.4±2.7 | 6.3±7.7 | 0.9±0.8 | 3.0±1.0 | 9.5±4.8 |
| m-Xylene | 0.4±0.2 | 1.2±1.2 | 0.1±0.1 | 0.4±0.2 | 0.3±0.1 | 17.2±9.0 | 53.8±56.1 | 3.6±2.6 | 16.7±8.0 | 12.5±5.9 |
| Styrene | 0.5±0.4 | 0.8±1.1 | 0.1±0.0 | 0.3±0.3 | 0.6±0.4 | 3.7±3.4 | 6.8±8.5 | 0.6±0.3 | 2.3±2.5 | 4.6±3.0 |
| o-Xylene | 0.2±0.1 | 0.7±0.8 | 0.1±0.0 | 0.1±0.1 | 1.5±2.2 | 6.0±3.9 | 23.6±29.3 | 2.1±1.4 | 5.1±2.4 | 53.2±79.4 |
| Isopropylbenzene | 0.0±0.1 | 0.4±0.5 | 0.0±0.1 | 0.0±0.1 | 0.2±0.1 | 0.2±0.4 | 5.3±6.9 | 0.2±0.3 | 0.1±0.2 | 2.8±1.7 |
| n-Propylbenzene | 0.0±0.1 | 0.4±0.5 | 0.0±0.1 | 0.0±0.1 | 0.0±0.1 | 0.5±0.4 | 4.0±5.3 | 0.1±0.3 | 0.2±0.4 | 0.0±0.1 |
| m-Ethyltoluene | 0.1±0.1 | 0.0±0.1 | 0.0±0.1 | 0.0±0.1 | 0.0±0.1 | 4.1±1.8 | 0.0±0.1 | 1.0±1.2 | 1.0±1.5 | 0.3±0.6 |
| p-Ethyltoluene | 0.0±0.1 | 0.3±0.3 | 0.0±0.1 | 0.0±0.1 | 0.1±0.1 | 1.0±0.6 | 7.1±6.6 | 0.3±0.5 | 0.0±0.1 | 1.3±1.8 |
| 1,3,5-Trimethylbenzene | 0.0±0.1 | 0.2±0.2 | 0.0±0.1 | 0.0±0.1 | 0.0±0.1 | 2.6±2.3 | 13.6±15.3 | 1.2±2.0 | 0.0±0.1 | 0.0±0.1 |
| 1,2,4-Trimethylbenzene | 0.1±0.1 | 0.3±0.3 | 0.2±0.1 | 0.0±0.1 | 0.0±0.1 | 2.6±3.1 | 13.7±13.4 | 7.4±3.6 | 2.3±2.4 | 0.2±0.7 |
| 1,2,3-Trimethylbenzene | 0.0±0.1 | 0.3±0.3 | 0.0±0.1 | 0.0±0.1 | 0.1±0.1 | 0.0±0.0 | 18.1±17.2 | 3.1±2.1 | 0.0±0.1 | 6.5±4.9 |
| o-Ethyltoluene | 0.0±0.1 | 0.2±0.2 | 0.0±0.1 | 0.0±0.1 | 0.0±0.1 | 1.2±1.4 | 7.0±7.0 | 0.8±0.8 | 0.6±0.9 | 0.7±0.9 |
| m-Diethylbenzene | 0.1±0.1 | 0.2±0.2 | 0.0±0.1 | 0.0±0.1 | 0.1±0.1 | 6.0±2.6 | 7.2±10.2 | 1.1±1.5 | 0.0±0.1 | 2.4±2.9 |

| Species/Groups | VOC mixing ratios (mean±std) (ppb) | | | | | OFP (mean±std) ($\mu g\ m^{-3}$) | | | | |
|---|---|---|---|---|---|---|---|---|---|---|
| | CD | CQ | TZ | XD | ZL | CD | CQ | TZ | XD | ZL |
| p-Diethylbenzene | 0.1±0.1 | 0.2±0.2 | 0.0±0.1 | 0.0±0.1 | 0.1±0.1 | 2.0±1.4 | 4.1±5.5 | 0.3±0.5 | 0.0±0.1 | 2.7±1.1 |
| Ethylene | 1.4±1.2 | 1.8±0.9 | 1.6±1.2 | 1.4±0.6 | 2.7±1.4 | 15.5±13.6 | 20.4±9.6 | 17.9±13.1 | 15.3±6.7 | 29.9±15.3 |
| Propylene | 0.4±0.4 | 0.9±0.7 | 0.7±0.5 | 0.6±0.4 | 1.6±1.5 | 8.8±8.0 | 20.7±15.6 | 14.6±11.8 | 12.5±8.2 | 36.0±33.9 |
| trans-2-Butene | 0.1±0.1 | 0.0±0.1 | 0.0±0.1 | 0.2±0.2 | 0.4±0.2 | 2.0±2.4 | 0.0±0.1 | 1.6±2.5 | 6.5±7.8 | 14.0±8.5 |
| 1-Butene | 0.0±0.1 | 0.0±0.1 | 0.0±0.1 | 0.1±0.2 | 0.2±0.2 | 1.2±1.6 | 1.0±1.1 | 1.0±1.7 | 3.2±4.1 | 5.0±3.7 |
| cis-2-Butene | 0.0±0.1 | 0.3±0.2 | 0.0±0.1 | 0.1±0.1 | 0.1±0.1 | 0.7±1.9 | 10.2±8.0 | 0.3±0.8 | 3.7±4.8 | 2.3±4.0 |
| trans-2-Pentene | 0.0±0.1 | 0.0±0.1 | 0.1±0.1 | 0.0±0.1 | 0.0±0.1 | 0.7±1.5 | 1.0±1.3 | 2.0±2.5 | 0.1±0.3 | 0.0±0.1 |
| 1-Pentene | 0.0±0.1 | 0.1±0.1 | 0.0±0.1 | 0.0±0.1 | 0.2±0.2 | 0.2±0.7 | 1.8±1.3 | 0.9±1.6 | 0.5±0.6 | 3.9±4.8 |
| cis-2-Pentene | 0.0±0.1 | 0.1±0.1 | 0.3±0.3 | 0.0±0.1 | 0.0±0.1 | 0.0±0.1 | 1.9±3.2 | 10.6±9.5 | 0.3±0.9 | 0.6±1.4 |
| 1-Hexene | 1.4±1.0 | 0.0±0.1 | 0.2±0.2 | 0.4±0.7 | 0.5±0.4 | 28.1±20.3 | 0.4±0.6 | 3.2±3.5 | 8.4±13.7 | 10.1±8.9 |
| Ethane | 4.0±1.5 | 3.3±1.6 | 4.4±3.9 | 2.4±0.8 | 0.0±0.1 | 1.5±0.6 | 1.2±0.6 | 1.7±1.5 | 0.9±0.3 | 0.0±0.1 |
| Propane | 2.8±1.7 | 3.9±2.4 | 13.3±11.1 | 4.1±2.3 | 3.2±0.8 | 2.7±1.6 | 3.7±2.3 | 12.8±10.7 | 4.0±2.2 | 3.1±0.8 |
| Isobutane | 0.8±0.6 | 1.4±1.4 | 1.0±1.2 | 1.3±1.1 | 3.2±1.4 | 2.6±1.9 | 4.5±4.6 | 3.2±3.9 | 4.1±3.4 | 10.3±4.5 |
| n-Butane | 1.6±0.9 | 2.4±2.4 | 4.4±3.6 | 2.9±2.5 | 0.9±0.5 | 4.7±2.8 | 7.0±7.2 | 13.0±10.9 | 8.7±7.6 | 2.7±1.6 |
| Cyclopentane | 0.1±0.1 | 0.3±0.2 | 2.5±2.2 | 0.3±0.5 | 1.6±0.9 | 0.3±0.3 | 0.9±0.8 | 8.9±7.8 | 1.2±1.7 | 5.9±3.1 |
| Isopentane | 1.3±0.7 | 0.4±0.8 | 0.0±0.1 | 8.6±5.0 | 0.1±0.1 | 10.3±5.5 | 3.4±5.9 | 0.0±0.1 | 66.2±38.5 | 0.6±0.5 |
| n-Pentane | 0.7±0.6 | 0.3±0.5 | 0.0±0.1 | 1.3±1.2 | 1.4±0.8 | 3.1±2.4 | 1.4±2.3 | 0.0±0.1 | 5.5±5.0 | 6.0±3.2 |
| 2,2-Dimethylbutae | 0.1±0.1 | 0.1±0.1 | 0.0±0.1 | 0.0±0.1 | 0.8±0.6 | 0.6±0.6 | 0.4±0.6 | 0.2±0.5 | 0.0±0.1 | 3.8±2.7 |
| 2,3-Dimethylbutane | 0.1±0.2 | 0.0±0.1 | 0.2±0.2 | 0.0±0.1 | 0.0±0.1 | 0.5±0.6 | 0.1±0.1 | 0.9±0.8 | 0.1±0.2 | 0.1±0.1 |
| 2-Methylpentane | 0.2±0.1 | 0.2±0.2 | 0.0±0.1 | 0.4±0.3 | 0.4±0.2 | 0.9±0.8 | 1.0±0.9 | 0.0±0.1 | 2.4±1.8 | 2.0±0.9 |
| 3-Methylpentane | 0.3±0.2 | 0.3±0.2 | 0.0±0.1 | 0.3±0.1 | 0.3±0.2 | 1.9±1.4 | 2.0±1.6 | 0.0±0.1 | 2.3±1.0 | 2.0±1.1 |
| n-Hexane | 0.6±0.4 | 0.2±0.3 | 0.0±0.1 | 0.6±0.2 | 0.1±0.1 | 2.7±2.0 | 0.9±1.4 | 0.1±0.4 | 2.7±1.1 | 0.4±0.4 |

| Species/Groups | VOC mixing ratios (mean±std) (ppb) | | | | | OFP (mean±std) (µg m$^{-3}$) | | | | |
|---|---|---|---|---|---|---|---|---|---|---|
| | CD | CQ | TZ | XD | ZL | CD | CQ | TZ | XD | ZL |
| Methylcyclopentane | 0.1±0.1 | 0.1±0.2 | 0.0±0.0 | 0.0±0.1 | 0.7±0.5 | 0.9±0.9 | 0.9±1.3 | 0.0±0.1 | 0.1±0.5 | 5.5±3.8 |
| 2,4-Dimethylpentane | 0.1±0.1 | 0.0±0.1 | 0.2±0.2 | 0.2±0.1 | 0.1±0.1 | 0.8±0.4 | 0.2±0.3 | 1.2±1.3 | 1.2±0.9 | 1.0±1.0 |
| Cyclohexane | 0.1±0.1 | 0.2±0.3 | 0.0±0.1 | 0.1±0.1 | 0.2±0.1 | 0.6±0.3 | 1.1±1.3 | 0.0±0.1 | 0.4±0.4 | 0.9±0.4 |
| 2-Methylhexane | 0.0±0.1 | 0.0±0.1 | 0.0±0.1 | 0.1±0.1 | 0.0±0.1 | 0.2±0.3 | 0.3±0.3 | 0.0±0.1 | 0.3±0.3 | 0.2±0.3 |
| 3-Methylhexane | 0.1±0.1 | 0.3±0.3 | 0.0±0.1 | 0.1±0.1 | 0.0±0.1 | 0.7±0.5 | 2.1±2.4 | 0.0±0.1 | 0.7±0.6 | 0.1±0.1 |
| 2,3-Dimethylpentane | 0.0±0.1 | 0.1±0.1 | 0.1±0.1 | 0.1±0.1 | 0.1±0.1 | 0.0±0.1 | 0.6±0.7 | 0.8±0.8 | 0.4±0.5 | 0.7±0.3 |
| 2,2,4-Trimethylpentane | 0.0±0.1 | 0.1±0.1 | 2.5±1.7 | 0.0±0.1 | 0.0±0.1 | 0.1±0.2 | 0.9±0.7 | 16.4±10.7 | 0.2±0.3 | 0.0±0.1 |
| n-Heptane | 0.1±0.1 | 0.4±0.3 | 0.6±0.4 | 0.2±0.1 | 0.0±0.1 | 0.4±0.4 | 2.1±1.6 | 2.8±2.1 | 0.8±0.4 | 0.2±0.2 |
| Methylcyclohexane | 0.0±0.1 | 0.2±0.2 | 0.0±0.1 | 0.1±0.1 | 0.1±0.1 | 0.2±0.4 | 1.3±1.5 | 0.2±0.5 | 1.0±0.9 | 0.7±0.6 |
| 2,3,4-Trimethylpentane | 0.0±0.1 | 0.2±0.2 | 0.2±0.4 | 0.0±0.1 | 0.0±0.1 | 0.0±0.1 | 1.2±1.2 | 1.2±2.0 | 0.0±0.1 | 0.0±0.1 |
| 2-Methylheptane | 0.0±0.1 | 0.2±0.2 | 0.0±0.1 | 0.0±0.1 | 0.1±0.1 | 0.0±0.1 | 1.1±1.1 | 0.2±0.5 | 0.1±0.1 | 0.3±0.4 |
| 3-Methylheptane | 0.0±0.1 | 0.8±0.5 | 0.2±0.2 | 0.0±0.1 | 0.0±0.1 | 0.2±0.3 | 4.9±3.2 | 1.5±1.3 | 0.1±0.2 | 0.0±0.1 |
| n-Octane | 0.0±0.1 | 0.3±0.3 | 0.1±0.1 | 0.0±0.1 | 0.0±0.1 | 0.1±0.1 | 1.2±1.4 | 0.4±0.5 | 0.2±0.2 | 0.1±0.2 |
| n-Nonane | 0.0±0.1 | 0.2±0.2 | 0.0±0.1 | 0.0±0.1 | 0.0±0.1 | 0.1±0.1 | 0.9±1.0 | 0.1±0.3 | 0.2±0.2 | 0.1±0.2 |
| n-Decane | 0.0±0.1 | 0.2±0.3 | 0.0±0.1 | 0.0±0.1 | 0.0±0.1 | 0.0±0.1 | 0.9±1.4 | 0.1±0.1 | 0.0±0.1 | 0.1±0.1 |
| n-Undecane | 0.0±0.1 | 0.2±0.2 | 0.1±0.0 | 0.0±0.1 | 0.0±0.1 | 0.1±0.1 | 0.8±0.9 | 0.3±0.2 | 0.1±0.2 | 0.0±0.1 |
| n-Dodecane | 0.0±0.1 | 0.2±0.2 | 0.1±0.1 | 0.0±0.1 | 0.0±0.1 | 0.0±0.1 | 0.8±0.9 | 0.3±0.2 | 0.0±0.0 | 0.0±0.1 |
| Formaldehyde | 4.5±2.3 | 6.6±3.5 | 15.9±20.0 | 8.1±4.3 | 7.8±3.8 | 56.6±29.1 | 83.3±44.4 | 202.0±253.8 | 103.2±54.8 | 99.1±48.0 |
| Acetaldehyde | 2.1±1.7 | 3.1±2.1 | 2.4±1.1 | 4.3±3.4 | 2.1±0.7 | 27.6±21.4 | 39.4±27.5 | 31.0±14.7 | 55.6±44.2 | 27.4±8.8 |
| Acetone | 2.6±1.5 | 2.9±1.8 | 1.9±1.2 | 2.6±1.0 | 2.7±1.6 | 2.4±1.4 | 2.7±1.6 | 1.8±1.2 | 2.4±0.9 | 2.5±1.5 |
| Acrolein | 0.0±0.1 | 0.0±0.1 | 0.0±0.1 | 0.1±0.2 | 0.0±0.1 | 0.0±0.1 | 0.2±0.6 | 0.0±0.2 | 1.4±4.6 | 0.1±0.3 |
| Propionaldehyde | 0.3±0.1 | 0.4±0.2 | 0.3±0.1 | 0.4±0.3 | 0.3±0.1 | 5.0±2.4 | 7.3±3.9 | 5.5±2.1 | 7.5±5.2 | 5.6±2.2 |
| Crotonaldehyde | 0.1±0.1 | 0.1±0.2 | 0.0±0.1 | 0.1±0.1 | 0.1±0.2 | 1.8±3.1 | 2.4±5.9 | 0.2±0.6 | 1.8±2.8 | 3.9±4.9 |
| Butyraldehyde | 0.2±0.1 | 0.2±0.3 | 0.2±0.1 | 0.8±0.9 | 0.1±0.1 | 3.5±2.6 | 4.7±5.2 | 3.4±2.3 | 14.7±16.7 | 2.8±2.4 |

| Species/Groups | VOC mixing ratios (mean±std) (ppb) | | | | | OFP (mean±std) ($\mu g\ m^{-3}$) | | | | |
|---|---|---|---|---|---|---|---|---|---|---|
| | CD | CQ | TZ | XD | ZL | CD | CQ | TZ | XD | ZL |
| Benzaldehyde | 0.1±0.1 | 0.2±0.1 | 0.2±0.1 | 0.3±0.6 | 0.2±0.1 | -0.4±0.4 | -0.8±0.4 | -0.5±0.3 | -1.0±1.9 | -0.6±0.4 |
| Cyclohexanone | 0.0±0.1 | 0.1±0.2 | 0.1±0.1 | 0.1±0.1 | 0.2±0.2 | 0.1±0.2 | 0.9±0.9 | 0.7±0.6 | 0.5±0.4 | 1.1±1.2 |
| Isovaleraldehyde | 0.1±0.2 | 0.1±0.1 | 0.1±0.2 | 0.1±0.1 | 0.0±0.1 | 1.9±4.0 | 1.1±1.1 | 2.5±4.1 | 1.2±0.6 | 0.9±1.0 |
| Pentanal | 0.0±0.1 | 0.0±0.1 | 0.1±0.1 | 0.1±0.1 | 0.1±0.1 | 0.5±0.8 | 0.7±0.8 | 1.8±1.2 | 1.3±0.7 | 1.6±1.4 |
| o-Tolualdehyde | 0.0±0.1 | 0.0±0.1 | 0.0±0.1 | 0.0±0.1 | 0.0±0.1 | 0.0±0.0 | 0.0±0.0 | 0.0±0.0 | 0.0±0.0 | 0.0±0.0 |
| m-Tolualdehyde | 0.1±0.1 | 0.1±0.1 | 0.1±0.1 | 0.1±0.1 | 0.1±0.1 | 0.0±0.0 | 0.0±0.0 | 0.0±0.0 | 0.0±0.0 | 0.0±0.0 |
| Hexaldehyde | 0.1±0.2 | 0.2±0.2 | 0.3±0.2 | 0.3±0.2 | 0.2±0.2 | 2.7±3.0 | 2.9±4.0 | 4.9±3.2 | 4.9±3.3 | 3.5±3.7 |
| Heptaldehyde | 0.0±0.1 | 0.0±0.1 | 0.1±0.1 | 0.0±0.1 | 0.0±0.1 | 0.0±0.1 | 0.2±0.5 | 1.3±1.0 | 0.3±0.5 | 0.2±0.4 |
| Octanal | 0.0±0.1 | 0.1±0.1 | 0.1±0.1 | 0.1±0.1 | 0.1±0.1 | 0.6±1.3 | 1.4±1.3 | 1.6±1.4 | 1.2±1.0 | 1.3±1.4 |
| Nonanal | 0.3±0.2 | 0.3±0.1 | 0.3±0.3 | 0.3±0.1 | 0.3±0.1 | 0.0±0.0 | 0.0±0.0 | 0.0±0.0 | 0.0±0.0 | 0.0±0.0 |
| Decanal | 0.2±0.1 | 0.1±0.1 | 0.1±0.1 | 0.2±0.1 | 0.2±0.1 | 0.0±0.0 | 0.0±0.0 | 0.0±0.0 | 0.0±0.0 | 0.0±0.0 |

15.  Figure 4. There is a typo on the y axis title

Response: Thank you for pointing it out. We have corrected the typo in the y-axis caption of

Figure 4 (b) and adjusted Figure 4.

[Figure]

Figure 4. (a) Ozone formation potential (OFP) and (b) proportions of OFP contributed by VOC subgroups, along with (c) mixing ratios of VOC subgroups at five sites.

16.  Line 296. Define acronym AOC and how this value is determined in the methods

Response: Thanks for point this out. We have added the definition of the acronym AOC and

explained how this value is calculated in the Methodology Section 2.4. Please refer to Lines 241-

248:

"Atmospheric oxidation capacity (AOC) is the core driving force of complex air pollution,

influencing the removal rate of trace gases and the production rates of secondary pollutants (Liu et

al., 2021). AOC is calculated based on the sum of oxidation rates of oxidants (OH, $O_3$, and $NO_3$)

with primary pollutants (VOCs, CO, and $CH_4$) (Elshorbany et al., 2009; Geyer et al., 2001; Yang et

al., 2022b). The formula is as follows:

$$AOC = \sum_i k_{Yi}[Y_i][X] \tag{7}$$

where $Y_i$ represents primary VOCs (excluding OVOCs), CO and CH₄, $X$ represents oxidants (OH, O₃ and NO₃) and $k_{Yi}$ is the bimolecular rate constant for the reaction of $Y_i$ with $X$. Atmospheric oxidation capacity determines the rate of $Y_i$ removal."

17. Fig. 7 &8. What is b) representing?

Response: Thank you for your question. In Figure 7 and Figure 8, panel (b) represents the difference of corresponding sources and sinks of $RO_x$ and $O_3$ between the Free and Base scenarios, respectively, illustrating the impact of OVOCs observationally constrained on the respective budgets. Specifically, in Figure 7(b), it shows the impact on the $RO_x$ budget, while in Figure 8(b), it shows the impact on the $O_3$ budget. Figure 7 and 8 have been moved to Figures 8 and 9, respectively.

18. Line 318-319. Needed acronyms definition for $O_3$ formation, $P(O_3)$, $L(O_3)$ and how these values are determined.

Response: Thank you for pointing it out. We have added definitions and calculation methods in Section 2.3 of the Methodology, Lines 219-232:

"Considering the oxidation of NO to NO₂ by peroxyl radicals, the total oxidant ($O_x = O_3 + NO_2$) is generally used to characterize the chemical budget of O₃ (Kanaya et al., 2009; Xue et al., 2014b). The total chemical production of $O_x$ through oxidations of NO by HO₂ and RO₂ radicals (Tan et al., 2018b), is defined as the production of O₃ (P(O₃)), which is calculated according to Equation (4):

$$P(O_3) = k_{HO2+NO}[HO_2][NO] + \sum k_{RO2,j+NO}[RO_2]_j[NO] \qquad (4)$$

The chemical loss rate (L(O₃)) of O₃ is equal to the sum of loss rates of O₃ and NO₂, including O₃ photolysis, reactions of O₃ with OH, HO₂ and alkenes, as well as reactions of NO₂ with OH and RO₂, as well as the reaction of NO₃ with unsaturated VOCs (Chen et al., 2020; Liu et al., 2022; Xue et al., 2016, 2014b).

$$L(O_3) = k_{O1D+H2O}[O1D][H_2O] + k_{O3+OH}[O_3][OH]$$
$$+ k_{O3+HO2}[O_3][HO_2] + k_{O3+alkenes}[O_3][alkenes]$$
$$+ k_{NO2+OH}[NO_2][OH] \qquad (5)$$
$$+ \sum k_{NO2+RO2,j}[NO_2][RO_2]_j$$
$$+ \sum k_{NO3+VOC,i}[NO_3][VOC]_i$$

The concentrations of radicals and intermediates are derived from the outputs of the OBM. The $k$ values in Equations (4) and (5) rate constants of the corresponding reactions, which can be found from https://mcm.york.ac.uk/MCM/ (last access: 13 Jan 2025) or the study by Liu et al. (2022). The subscript '$j$' in Equation (4) and (5) denotes individual RO₂ species. The subscript '$i$' in Equation (5) represents individual VOC species. The net O₃ production rate can be obtained from the difference between P (O₃) and L(O₃)."

19. Line 339. Conducted in China or urban areas in general? What is the upper-middle range? A review of the existing studies would be helpful (if the review is the same presented in the SI, I suggest to include the review in the main text).

Response: Thank you for your comment. The previous studies were conducted in China (see Table S4). The upper-middle range refers to the range of VOC mixing ratios above the median value

(> 32 ppb), as determined from the literature review and data collection presented Table S4 in the

SI. We have added a bar chart comparing VOCs by city in the main text for better clarity. Please

refer to Figure 3.

[Figure]

**Figure 3. Comparison of VOC mixing ratios and compositions in this study with former studies based on Table S4. The red dash line represents the median levels (~32 ppb) of VOCs。**

20. Line 348. Refer to SI where needed.

Response: Thanks for your suggestion. We have inserted the relevant chart to support the

information, and have referred to Table S5 in the supporting information. Please refer to specific

comment 14.

21. Line 356. Can the authors separate emissions from transport? For example using an auxiliary

method, such as a source apportionment method or ancillary measurements

Response: Thank you for the insightful comment regarding the separation of OVOC emissions from transport. The OBM is designed to simulate in-situ photochemical production, chemical loss, and deposition processes within a given air mass at a specific location. It does not explicitly account for emissions or regional transport processes (Wolfe et al., 2016), so emissions and transport were combined into a single term ($R_{Emis\&Trans}$) in our study. However, after many attempts, we apologize that we were unable to modify the emission and transmission contributions to OVOC. We would like to elaborate on the reasons for this.

While ancillary methods such as source assignment (e.g., the positive matrix factorization (PMF) model (Bon et al., 2011), the multiple linear regression (MLR) (Garcia et al., 2006; Zou et al., 2024), and the photochemical age-based parameterization method) (de Gouw et al., 2005; Huang et al., 2020), can theoretically help to isolate emissions during transportation, their application in separating transport and emissions was limited by the following factors:

1) Tracer representativeness:

    (1) Methods like MLR and PMF rely on tracer species to characterize sources. For example, CO, benzene, and toluene are commonly used as tracers for anthropogenic emissions (Huang et al., 2019; Zou et al., 2024). However, identifying tracers that specifically represent transport processes is challenging, as transport-influenced air masses often lack unique chemical markers.

    (2) In the photochemical age-based parameterization approach, the photochemical age is used to characterize aged air masses (de Gouw et al., 2005). However, this is to a large extent closely related to the secondary photochemical reactions, which does not allow

a better quantification of the contribution of transport.

2) Complexity of urban environments:

(1) Photochemical age‑based parameterization methods require a fixed emission ratio of each VOC relative to a selected tracer, which is often unavailable or highly variable in complex urban environments (Yuan et al., 2012; Zou et al., 2024).

(2) These source apportionment methods also struggle to distinguish between primary and secondary sources, particularly when specific precursors are not well-defined or when multiple sources contribute to OVOCs.

Given these limitations, auxiliary methods are not well-suited to separate emissions from transport in the context of our study. We acknowledge that separating emissions from transport is crucial for a more comprehensive understanding of OVOCs sources and sinks. Integrating the OBM with regional chemical transport model (CTM) to explicitly may resolve emissions and transport dynamics.

---

## Author Comment (AC2)

**Response to Reviewers' comments:**

**Reviewer #1**: The article "Significant influence of oxygenated volatile organic compounds on atmospheric chemistry analysis: A case study in a typical industrial city in China" highlights OVOCs as critical players for air quality analysis. Dai et al. base their study on an experimental dataset of VOC, OVOC and other air quality and meteorological parameters at five monitoring sites within the city of Zibo. They present these data and analyse the role of OVOC on radical chemistry and ozone formation potential with an observation-based model (OBM). With this, the study fits well into the scope of ACP. The combined approach of experimental and theoretical analysis is interesting and valid for the study's overall aim of comparing the different sites and the impact of including or excluding OVOC on a typical air quality analysis and conclusions for ozone production potential.

After reading the preprint, I have three major critiques, which I want to outline here and later on discuss in the specific comments in further detail.

Response: We thank the reviewer for the detailed and constructive comments on our manuscript. Below is our point-by-point response to each comment, marked in blue. Changes made to the main text are marked in blue in the revised manuscript file.

**General comments:**

Firstly, the study is based on the comparison of five experimental datasets from five different sites in the city of Zibo, which were monitored in parallel during August 8-12, 2021. VOC were measured with online GC-FID and OVOC were sampled on sorbent tubes and analysed offline with HPLC. However, I neither see a list of the measured VOC or OVOC nor information about calibrations and uncertainties. Most importantly, how do you know that the results of the five stations are comparable? Did you perform any comparative side-by-side measurements?

Response: We appreciate your valuable feedback. The VOC analyzing instruments used at all five sites were operated by the same organization and underwent regular calibration with same standard gas. For OVOCs, offline sampling at all sites was conducted using identical sampling instruments, methods, preservation techniques, and analytical procedures. Therefore, we believe that the observations from the

five sites are comparable. Additionally, a detailed list of the VOCs and OVOCs measured at each site

has been provided in the supplementary materials, please see Table S2 for details.

Table S2 VOCs species and their names in Master Chemical Mechanism (MCMv3.3.1), minimum detection limits (MDL), and maximum incremental reactivity coefficient (MIR). "—" means that the species is not listed in the mechanism.

| Species | MCM name | MDL (ppb) | MIR | Species | MCM name | MDL (ppb) | MIR |
|---|---|---|---|---|---|---|---|
| **Alkanes** | | | | **BVOCs** | | | |
| Ethane | C2H6 | 0.079 | 0.28 | Isoprene | C5H8 | 0.02 | 10.61 |
| Propane | C3H8 | 0.046 | 0.49 | **Alkynes** | | | |
| Isobutane | IC4H10 | 0.022 | 1.23 | Acetylene | C2H2 | 0.032 | 0.95 |
| n-Butane | NC4H10 | 0.027 | 1.15 | **Aromatics** | | | |
| Cyclopentane | — | 0.016 | 1.15 | Benzene | BENZENE | 0.012 | 0.72 |
| Isopentane | IC5H12 | 0.087 | 2.39 | Toluene | TOLUENE | 0.013 | 4.00 |
| n-Pentane | NC5H12 | 0.031 | 1.31 | Ethylbenzene | EBENZ | 0.014 | 3.04 |
| 2,2-Dimethylbutae | M22C4 | 0.014 | 1.17 | m-Xylene | MXYL | 0.027 | 9.75 |
| 2,3-Dimethylbutane | M23C4 | 0.019 | 0.97 | Styrene | STYRENE | 0.014 | 1.73 |
| 2-Methylpentane | M2PE | 0.031 | 1.50 | o-Xylene | OXYL | 0.012 | 7.64 |
| 3-Methylpentane | M3PE | 0.012 | 1.80 | Isopropylbenzene | IPBENZ | 0.014 | 2.52 |
| n-Hexane | NC6H14 | 0.011 | 1.24 | n-Propylbenzene | PBENZ | 0.013 | 2.03 |
| Methylcyclopentane | — | 0.011 | 2.19 | m-Ethyltoluene | METHTOL | 0.032 | 7.39 |
| 2,4-Dimethylpentane | — | 0.013 | 1.55 | p-Ethyltoluene | PETHTOL | 0.014 | 4.44 |
| Cyclohexane | CHEX | 0.016 | 1.25 | 1,3,5-Trimethylbenzene | TM135B | 0.012 | 11.76 |
| 2-Methylhexane | M2HEX | 0.012 | 1.19 | 1,2,4-Trimethylbenzene | TM124B | 0.011 | 8.87 |
| 3-Methylhexane | M3HEX | 0.013 | 1.61 | 1,2,3-Trimethylbenzene | TM123B | 0.011 | 11.97 |
| 2,3-Dimethylpentane | — | 0.013 | 1.34 | o-Ethyltoluene | OETHTOL | 0.013 | 5.59 |
| 2,2,4-Trimethylpentane | — | 0.012 | 1.26 | m-Diethylbenzene | — | 0.011 | 7.10 |
| n-Heptane | NC7H16 | 0.012 | 1.07 | p-Diethylbenzene | — | 0.011 | 4.43 |
| Methylcyclohexane | — | 0.011 | 1.70 | **OVOCs** | | | |
| 2,3,4-Trimethylpentane | — | 0.013 | 1.03 | Formaldehyde | HCHO | 0.007 | 9.46 |
| 2-Methylheptane | — | 0.013 | 1.07 | Acetaldehyde | CH3CHO | 0.016 | 6.54 |

| | | | | | | | |
|---|---|---|---|---|---|---|---|
| 3-Methylheptane | — | 0.013 | 1.24 | Acetone | CH3COCH3 | 0.009 | 0.36 |
| n-Octane | NC8H18 | 0.012 | 0.90 | Acrolein | ACR | 0.008 | 7.45 |
| n-Nonane | NC9H20 | 0.013 | 0.78 | Propionaldehyde | C2H5CHO | 0.026 | 7.08 |
| n-Decane | NC10H22 | 0.011 | 0.68 | Crotonaldehyde | C4ALDB | 0.042 | 9.39 |
| n-Undecane | NC11H24 | 0.018 | 0.61 | Butyraldehyde | C3H7CHO | 0.048 | 5.97 |
| n-Dodecane | NC12H26 | 0.048 | 0.55 | Benzaldehyde | BENZAL | 0.055 | -0.67 |
| **Alkenes** | | | | Cyclohexanone | CYHEXONE | 0.058 | 1.35 |
| Ethylene | C2H4 | 0.057 | 9.00 | 3-Methylbutyraldehyde | C3ME3CHO | 0.058 | 4.97 |
| Propylene | C3H6 | 0.022 | 11.66 | Pentanal | C4H9CHO | 0.038 | 5.08 |
| trans-2-Butene | TBUT2ENE | 0.013 | 15.16 | o-Tolualdehyde | OXYLAL | 0.072 | 0.00 |
| 1-Butene | BUT1ENE | 0.023 | 9.73 | m-Tolualdehyde | MXYLAL | 0.089 | 0.00 |
| cis-2-Butene | CBUT2ENE | 0.016 | 14.24 | Hexaldehyde | C5H11CHO | 0.060 | 4.35 |
| trans-2-Pentene | TPENT2ENE | 0.012 | 10.56 | Heptaldehyde | C6H13CHO | 0.034 | 3.69 |
| 1-Pentene | PENT1ENE | 0.093 | 7.21 | Octanal | — | 0.029 | 3.16 |
| cis-2-Pentene | CPENT2ENE | 0.011 | 10.38 | Nonanal | — | 0.032 | 0.00 |
| 1-Hexene | HEX1ENE | 0.014 | 5.49 | Decanal | — | 0.035 | 0.00 |

Secondly, the study is presented in three parts:

1. Observations and comparison of air composition at the five sites within the city of Zibo

2. OVOC sources and sinks

3. Importance of OVOC in the OBM air quality analysis

While I can follow the presentation of results and argumentation of the discussion for parts 1 and 3 very well, I am not convinced by the approach, presentation and discussion of part 2. As far as I could understand, the budget of sinks and sources in the box model is used to describe the time-dependent rate of change of the OVOC as observed with the measurements. The budget of sinks and sources consists of the net production (R_NetProd), the deposition (R_Dep), and emissions and transport (R_Emis&Trans). Here, I have several questions: Why were emissions and transport combined? Why is the chemical or photolytic loss of the OVOC not included in this budget? In order to understand the conclusions of part 2, I would need a much more precise description of the methods, chosen approach, and better references.

Response: Thanks for your thoughtful feedback. We apologize for any confusion or questions that may have arisen due to the unclear expression. For question 1, at a given location, OVOCs concentrations are influenced by chemical processes, emissions, transport, deposition, and dilute mixing (Tan et al., 2018b; Xue et al., 2014b; Zhang et al., 2021). The method for the separation of each process is based on the method of Xue et al., 2014b, which is originally proposed for the investigation of $O_3$.

$$R_{meas} = \frac{d(O_3)}{dt}$$

$$R_{chem} = P(O_3) - L(O_3)$$

$$R_{trans} = R_{meas} - R_{chem} - R_{dep}$$

In this formular, the measured $O_3$ concentrations can be divided into three parts $R_{chem}$, $R_{dep}$, and $R_{trans}$. Similarly, the OVOCs concentrations can also be divided into three parts, where $R_{NetProd}$, and $R_{Dep}$ are identically same as the definition of $R_{chem}$, and $R_{dep}$ in the study of Xue et al., 2014b. However, OVOCs can be directly emitted from surface source ($R_{Emis}$), which is different from $O_3$. According to

the method of Xue et al., 2014b, the sum of $R_{Trans}$ and $R_{Emis}$ should equal to $R_{meas}$ minus $R_{chem}$ and $R_{dep}$. However, the 0-D observation-based model has a limitation of not considering transport and emissions. Therefore, the emissions and transport are combined in our calculations.

For question 2, we have to apologize for the confusion that may have arisen due to the ambiguous introduction. In fact, $R_{NetProd}$ represents the combined effects of VOC oxidation producing OVOCs, OVOC photolysis, and reactions of OVOCs with oxidants (OH, $NO_3$, and $O_3$). Hence, the chemical or photolytic loss of the OVOC has been included in our analysis.

For question 3, at a given location, the variation of OVOCs mixing ratios is mainly influenced by in-situ photochemical production and chemical loss, emissions and regional transport, and deposition (Tan et al., 2018b; Xue et al., 2014b; Zhang et al., 2021). However, the 0-D box model (OBM) primarily account for atmospheric photochemical reactions, deposition, and dilution mixing within the boundary layer, without explicitly representing transport processes and emissions (Wolfe et al., 2016; Zhang et al., 2021). Therefore, emissions and transport are combined in our analysis of OVOC budget to account for their contributions collectively. To avoid misleading, the descriptions of the methodology for calculating the source and sink of OVOCs have been refined in Section 2.3, please refer to Lines 201-218:

"At a given site, variations in OVOCs mixing ratios are mainly influenced by in-situ photochemical production and chemical loss, emissions, regional transport, and deposition (Tan et al., 2018a; Xue et al., 2014a; Zhang et al., 2021). The change rate of observed OVOCs ($R_{Meas}$) is calculated by Equation (1). The in-situ photochemical production of OVOC ($R_{ChemProd}$) is mainly caused by the oxidation of VOCs, while their in-situ chemical loss ($R_{ChemLoss}$) includes photolysis and reactions with oxidants (OH, $NO_3$, and $O_3$) ([https://mcm.york.ac.uk/MCM/](https://mcm.york.ac.uk/MCM/), last access: 13 Jan 2025) (Atkinson, 2000; Atkinson and Arey, 2003; Jenkin et al., 2015; Saunders et al., 2003). The in-situ net OVOCs chemical production ($R_{NetProd}$) (Equation (2)) and their removal by deposition ($R_{Dep}$) are calculated hourly according to the OBM simulation. The OBM primarily accounts for atmospheric photochemical reactions, and deposition within the boundary layer. However, previous studies have reported that the OBM lacks an explicit representation of transport processes and emissions (Wolfe et al., 2016; Zhang et al., 2021), making it challenging to disentangle their respective contributions. Therefore, emissions and transport

are combined to a single term ($R_{Emis\&Trans}$) to represent their contributions collectively. If the $R_{Emis\&Trans}$ is positive, it is considered a net import of emissions/transport, whereas a negative suggests a net export. The emissions and regional transport of OVOCs ($R_{Emis\&Trans}$) are computed as Equation (3).

$$R_{Meas} = \sum_i \frac{d([OVOC]_i)}{dt}$$

(1.)

$$R_{NetProd} = \sum_i (R_{ChemProd,i} - R_{ChemLoss,i})$$

(2.)

$$R_{Emis\&Trans} = (R_{Meas} - R_{NetProd} - \sum_i R_{Dep,i})$$

(3.)

where $[OVOC]_i$ is the mixing ratios of OVOC species $i$ constrained in OBM, 15 in total (Table S2). dt is the time-step of modeling, $d[OVOC]_i$ refer to the change in mixing ratio of OVOC species $i$."

Thirdly, it seems like the model set-up and constrain were complex and based on several assumptions (e.g. as outlined in SI). I miss a thorough presentation of these in the manuscript and the discussion of the model uncertainties. When using the MCM mechanism for analysing the role of OVOC, the $RO_x$ chemistry and ozone formation potential (OFP), which draw-backs do you expect or which questions cannot be answered with your model approach? Where are the limitations?

Response: Thanks for the valuable comment. The detailed description model set-up and constrain and the uncertainty of the model is essential for understanding the reliability of the conclusion and the application of the results.

Firstly, a more detailed description of model set-up and constraints is provided in Section 2.2, please refer to Lines 161-199:

"A box model (F0AM) coupled with the Master Chemical Mechanism (MCM) v3.3.1 was utilized to simulate the in situ atmospheric chemical process at these 5 sites (Jenkin et al., 2015; Wolfe et al., 2016). The MCMv3.3.1, as a nearly explicit mechanism with more than 5800 species and 17000 reactions, provides a more detailed gas chemistry than other lumped mechanisms, such as the Carbon Bond Mechanism (CB) (Yarwood et al., 2005, 2010), Regional Atmospheric Chemistry Mechanism (RACM) (Goliff et al., 2013; Stockwell et al., 1997b), and SAPRC (Carter, 1990, 2010b; Carter and

Heo, 2013). The box model calculations were constrained by comprehensive measurements of trace gases (NO, $NO_2$, CO, and $SO_2$) and 45 speciated VOCs, encompassing 20 alkanes, 9 alkenes, 14 aromatics, 15 OVOCs, isoprene and acetylene, as well as meteorological parameters (T, RH and P). To address potential $NO_2$ measurement artifacts, several adjustments were implemented. Considering that PKU-Mo as a catalytic converter for $NO_2$ measurement can cause interferences from other nitrogen–oxygen compounds (e.g., PAN, $HNO_3$), potentially overestimating $NO_2$ by 30%~50% (Kim et al., 2015; Tan et al., 2017, 2019a; Xu et al., 2013). In this study, the observed $NO_2$ mixing ratio ($[NO_2]_{obs}$) at the 5 sites were reduced by 30%~40% (40% for ZL and CQ, 30% for CD, TZ and XD) to compensate for catalytic converter interferences (Xu et al., 2013). Additionally, strong anthropogenic emissions (e.g., vehicle emissions) near the sites may prevent the model from reaching steady state, leading to positive deviations (Li et al., 2014c). Therefore, NO steady-state approximations ($[NO]_{ss}$), calculated according to the equations proposed by Del Negro et al. (1999) (Equation S1), was used to constrain the simulated NO. The uncertainties derived from the $NO_x$ settings are shown in Table S3 and analyzed in Section 3.4. HONO was fixed to 2% of the corrected $NO_2$ mixing ratio ($[NO_2]_{cor}$) (Elshorbany et al., 2012; Tan et al., 2019a), and the corresponding uncertainty is summarized in Section 3.4. In addition, boundary layer height (BLH), and surface net solar radiation (SSR) were obtained from the fifth generation of the European Centre for Medium-Range Weather Forecasts (ECMWF) reanalysis for the global climate and weather (https://cds.climate.copernicus.eu, last access: March 1, 2024). The photolysis frequency correction factor (Jcorr) of the model input was adjusted by SSR. BLH was also included in the model to control the deposition process (Xuan et al., 2023; Zhu et al., 2020).

The model ran with continuous time series profile for the campaign period (August 8-12) with 1-hour time-step. A sensitivity analysis was performed for the time-step and the results are summarized in Section 3.4. Each simulation started with 10-days spin up to reach steady state condition. Missing observation data were filled with linear interpolation, and the mixing ratios of OVOCs were also linearly interpolated to 1-hour resolution for modeling. An artificial loss process corresponding to an atmospheric lifetime of 24 h or a first-order dilution rate (kdil) of 1/86400 $s^{-1}$ was introduced for all simulated species, including secondary species and radicals, to approximately simulate dry deposition

and other losses (Lou et al., 2010; Tan et al., 2018b; Wang et al., 2022c). The model cases that run with the above settings with 15 constrained OVOCs species are called the Base scenario. To investigate the impacts of constrains of OVOCs on atmospheric chemistry, the Free scenario was conducted, with all the setting of the Base scenario except for the OVOCs constraint."

Secondly, as described in Text S1, the uncertainty of the model mainly comes from the setting of $NO_x$. Sensitive model runs are performed with different $NO_x$ settings to show the corresponding uncertainty. The related descriptions have been added, please refer to Lines 446-459:

"The uncertainty of the model mainly comes from the setting of $NO_x$ (Text S1). Sensitive model runs are performed with different $NO_x$ settings to show the corresponding uncertainty. Firstly, $NO_{ss}$, $[NO_2]_{cor}$, VOCs, OVOCs, T, RH, and BLH are constrained in the Base scenario, and sensitivity simulations were carried out for gradient $NO_2$ mixing ratios (50%, 60%, 70% of $[NO_2]_{obs}$). Generally, different $NO_2$ settings produce uncertainties of -12%~7% for daytime $O_3$ and -30%~10% for the daytime $R_{NetProd}$ of OVOCs. In addition, different $NO_2$ settings produce uncertainties of -122%~14% for the $R_{Emis\&Trans}$ of OVOCs. This indicates that reducing the uncertainty of $NO_2$ observations is important for further atmospheric chemistry modelling. Secondly, the $[NO]_{ss}$ and observed NO mixing ratio ($[NO]_{obs}$) was respectively used in the model to investigate the influence of nearby NO emissions on the OBM result (Table S3). Sensitivity tests show that directly using the $[NO]_{obs}$ may resulted in an uncertainty of 3% ~20%, 10% ~27% and 7.1%~38% for $O_3$, $R_{NetProd}$ and $R_{Emis\&Trans}$, respectively, with significantly higher uncertainties at TZ than at other stations, which may be related to the strong transient emissions of $NO_x$."

Below is the equation used to calculate $NO_{ss}$, which is provided in the SI:

$$NO_2 + hv \xrightarrow{R_1} NO + O \qquad \text{R1}$$
$$NO_2 + hv \xrightarrow{R_2} NO + O \qquad \text{R2}$$
$$[NO]_{ss} = \frac{J_{NO2} * [NO_2]}{k_{NO+O3} * [O_3]} \qquad \text{(S1)}$$

Where $J_{NO2}$ represents the photolysis rate coefficient for reaction R1, $k_{NO+O3}$ represents the reaction rate coefficient for the reaction R2. $[NO_2]$, $[O_3]$ and $[NO]_{ss}$ represents the mixing ratios of $NO_2$, $O_3$ and steady-state approximations of NO, respectively.

**Table S1 Uncertainty in sensitive model runs performed with different $NO_x$ settings**

| Parameter | Site | $[NO_2]_{cor}$ [a] and $[NO]_{ss}$ [c] (the Base scenario) (ppb or ppb h$^{-1}$) [d] | Changes based on the Base scenario | | | |
|---|---|---|---|---|---|---|
| | | | $0.5*[NO_2]_{obs}$ [a] | $0.6*[NO_2]_{obs}$ | $0.7*[NO_2]_{obs}$ | $[NO_2]_{cor}$ and $NO_{obs}$ [b] |
| Daytime $O_3$ | CD | 60.9 | -6.9% | -2.8% | — | 5.3% |
| | CQ | 82.4 | -8.3% | — | 7.4% | 19.2% |
| | TZ | 66.2 | -15.3% | -7.3% | — | 8.4% |
| | XD | 85.2 | -12.1% | -5.6% | — | 2.9% |
| | ZL | 96.8 | -5.9% | — | 4.8% | 7.5% |
| Daytime $R_{NetProd}$ | CD | 3.5 | -10.2% | -4.4% | — | 8.7% |
| | CQ | 4.1 | -11.5% | — | 10.3% | 27.1% |
| | TZ | 1.9 | -30.9% | -14.7% | — | 17.0% |
| | XD | 3.6 | -17.2% | -7.9% | — | 4.4% |
| | ZL | 5.9 | -7.2% | — | 5.8% | 9.8% |
| $R_{Emis\&Trans}$ | CD | -1.3 | -13.6% | -5.9% | — | 10.8% |
| | CQ | -1.5 | -16.0% | — | 14.5% | 37.6% |
| | TZ | -0.2 | -122.4% | -58.1% | — | 67.2% |
| | XD | -1.2 | -26.9% | -12.5% | — | 7.1% |
| | ZL | -2.5 | -9.1% | — | 7.5% | 12.2% |

Note: [a] $[NO_2]_{obs}$ and $[NO_2]_{cor}$ represents the mixing ratios of observed and corrected $NO_2$. The $[NO_2]_{cor}$ of CD, TZ and XD are $0.7*[NO_2]_{obs}$, and those of CQ and ZL are $0.6*[NO_2]_{obs}$ (Text S1); [b] $[NO]_{obs}$ represents the mixing ratios of observed NO; [c] $[NO]_{ss}$ represents the steady-state approximations of NO mixing ratio; [d] The ppb is for $O_3$, and ppb h$^{-1}$ for $R_{NetProd}$ and $R_{Emis\&Trans}$.

Thirdly, while the OBM has advantages in modeling using a near-explicit master chemical mechanism (MCM) and reproducing atmospheric chemistry at a given location based on observations, it also has inherent limitations.

1) Inadequate consideration of emission/transport processes.

Measured VOC/NO$_x$ levels are affected by emissions, photochemistry, transport, and deposition. However, in constrained models using observed data, the OBM cannot explicitly represent emission and transport processes, which can create uncertainty in the simulation (Wolfe et al., 2016). This may make the modeling of atmospheric chemical analysis biased from realistic.

2) Limitations of the steady-state assumption.

When using observations to constrain the model, OBM calculations are based on the assumption of steady-state conditions. However, this assumption is invalid when the site is strongly influenced by nearby emissions (Wolfe et al., 2016). In such cases, the model might not adequately represent the actual atmospheric chemistry, potentially leading to an overestimation of secondary products, such as O$_3$, RO$_x$ radicals and OVOCs (Li et al., 2021, 2014b). Due to the high traffic density in Zibo, these sites are subject to strong local traffic emissions. This strong source influence violates the approximate steady-state assumption when using direct observations to constrain the OBM. As a solution, we calculated NO steady-state approximations (NO$_{ss}$) on the basis of corrected NO$_2$ to better represent realistic atmospheric conditions.

3) Missing or incomplete chemistry.

The model may not fully capture all relevant chemical processes, such as heterogeneous reactions or the formation and fate of intermediate species (Xue et al., 2014b; Yu et al., 2020, 2022). For example, the role of ClNO$_2$ in atmospheric chemistry is not included, which could lead to underestimation of RO$_x$ and O$_3$ production in certain environments (Riedel et al., 2014; Xia et al., 2020). Follow-up research may be able to gradually refine the chemical mechanism to approximate the actual atmosphere more closely.

These limitations of OBM have been summarized and addressed in the uncertainty part, please see Lines 485-499:

"The OBM, while advantageous for its use of the near-explicit master chemical mechanism and its ability to reproduce atmospheric chemistry based on observations, has several inherent limitations. Firstly, it inadequately considers emission and transport processes (Wolfe et al., 2016), leading to uncertainties and potential biases in atmospheric chemical analysis. Future studies could differentiate the two processes using CTMs. Secondly, the steady-state assumption, which underpins OBM calculations, might become invalid in areas strongly influenced by nearby emissions (e.g., high-traffic and industrial intensive regions) (Wolfe et al., 2016). This may potentially cause overestimations of secondary products such as $O_3$, $RO_x$ radicals, and OVOCs (Li et al., 2021, 2014b). In this study, simulations use $NO_{ss}$ as inputs, acknowledging that traffic-related NO emissions may prevent the system from reaching an approximate steady state. The associated uncertainties have been discussed above. Lastly, the model may not fully capture all relevant chemical processes, such as heterogeneous reactions or the formation and the fate of intermediate species. For instance, the exclusion of $ClNO_2$ chemistry could result in underestimating $RO_x$ and $O_3$ production in certain environments (Riedel et al., 2014; Xia et al., 2020). Future research should focus on refining the chemical mechanisms to better approximate real atmospheric conditions."

Finally, as Dai et al. contribute to the current discussion of VOC and OVOC for urban air quality, I would like to recommend this article for publication, yet, with major revisions. Please, find in the following my specific comments that should be addressed before publication.

**Specific comments:**

Abstract

1.      In general, the abstract would benefit from a more precise and scientific wording. Please, consider to ask a native speaker to review the manuscript for assistance in improving the English. In the abstract, it would be great to highlight the findings of each of the three results-parts more clearly.

Response: Thanks for your constructive suggestions, which will help us improve the manuscript significantly. We understand the importance of precise and scientific wording in the abstract and

appreciate your suggestion to involve a native speaker for improving the manuscript's language quality. We have carefully reviewed the abstract and revised it to ensure clarity and precision.

In particular, we have restructured the abstract to clearly outline the key findings of each of the three result sections. Please refer to Lines 12-31:

"Oxygenated volatile organic compounds (OVOCs), an important subgroup of volatile organic compounds (VOCs), are emitted directly or formed secondarily through photochemical processes. They play a crucial role in tropospheric chemistry as ozone ($O_3$) precursors. However, due to measurement limitations, the influence of OVOCs on $O_3$ formation has often been underestimated. In this study, 74 VOCs (including 18 OVOCs) were measured at five representative stations (urban, suburban, industrial, upwind, and downwind stations) in Zibo, an industrial city in the North China Plain. The VOCs level in Zibo (44.6±20.9 ppb) is in the upper-middle range (> 32 ppb) compared to previous studies conducted in most Chines cities, with OVOCs contributing for 30.0%~37.8%. The average $O_3$ formation potential in Zibo is 410.4±197.2 $\mu g \ m^{-3}$, with OVOCs being the dominant contributor (31.5%~55.9%). An observation-based model (OBM) was used to access the contributions of chemical production ($R_{NetProd}$) and emissions/transport ($R_{Emis\&Trans}$) to individual OVOC. Daytime (8:00-18:00 LT) $R_{NetProd}$ is the highest at the urban site (5.9 ppb $h^{-1}$), while nighttime $R_{Emis\&Trans}$ is most negative at the industrial site (0.76 ppb $h^{-1}$). Simulations without OVOCs constraint overestimate OVOCs (42.1~126.5%) and key free radicals (e.g., (hydroperoxy radicals, $HO_2$) (5.3%~20.4%) and (organic peroxy radicals, $RO_2$) (6.6%~35.1%)), leading to a 1.8%~11.9% $O_3$ overestimation. This overestimation causes an underestimation of hydroxyl radicals (OH) (1.8%~20.9%) and atmospheric oxidizing capacity (3.5%~12.5%). These findings emphasize the importance of comprehensive OVOCs measurements to constrain numerical models, especially in regions with dense anthropogenic emissions, to better reproduce atmospheric photochemistry, and to formulate more effective air pollution control strategies."

2.    Line 12-14: I stumbled upon the formulation in the first sentence as it seems rather bulky and unprecise. Please rephrase. For example, what do you mean with the statement that OVOC are "key components" of VOC? Do you want to highlight them as a sub-group of VOC? Please, also check for

English grammar, as I believe "secondary generated" is not correct. Why do OVOC play a crucial role in tropospheric chemistry? I think the sentence could work better, if you shortended it and added causality between this statement and the ozone precursors (e.g. …as they can act as ozone precursors).

Response: Thank you for your detailed feedback. We agree that the original sentence is bulky and could be more precise in its formulation. The related description has been revised, please refer to Lines 12-14:

"Oxygenated volatile organic compounds (OVOCs), an important subgroup of volatile organic compounds (VOCs), are emitted directly or formed secondarily through photochemical processes. They play a crucial role in tropospheric chemistry as ozone ($O_3$) precursors."

3. Line 18: What is the upper-middle range of VOC mixing ratios in previous studies? Which locations do you refer to?

Response: We apologize for the ambiguity regarding the "upper-middle range" of VOC mixing ratios. To clarify, the upper-middle range of VOC mixing ratios refers to levels higher than the median values (~32 ppb) derived from previous studies, as summarized in Table S3. These studies conducted in Chinese cities, offering a statistical basis for comparison.
The related description has been revised, please refer to Lines 18-19:

"The VOCs level in Zibo (44.6±20.9 ppb) is in the upper-middle range (> 32 ppb) compared to previous studies conducted in most Chines cities, with OVOCs contributing for 30.0%~37.8%."

4. Line 18: Here (and elsewhere in the text) you present numbers in % with one decimal place 30.0-37.8%. Considering the uncertainties of VOC and OVOC measurements and the width of this range, is it meaningful to report the one digit on the right of the decimal point?

Response: Thank you for your careful review and valuable feedback regarding our work. We appreciate your concern about the precision of the reported percentages. Given that the detection limits for VOC and OVOC measurements are typically <0.1 ppb, we believe that retaining one decimal place is appropriate.

5.         Line 19: The overall $O_3$ formation potential in Zibo… Do you mean the average $O_3$ formation potential?

Response: Thank you for your observation. Yes, when we refer to the "overall $O_3$ formation potential in Zibo…" we are indeed referring to the average $O_3$ formation potential across the five stations. We apologize for any confusion this may have caused.

6.         Line 21: Why did you combine emissions and transport in the assessment of contributions to OVOC?

Response: Thanks for your focused attention on this issue. We would like to clarify the reason behind combining emissions and transport in the assessment of contributions to OVOC. As responded to the second major critiques above, the changes of measured OVOCs mixing ratios are caused by chemical production/destruction and physical processes (deposition, horizontal/vertical mixing, and emissions). The 0-D box model only considers atmospheric photochemistry, deposition, and dilution mixing, without explicitly representing emission and transport processes. Due to model limitations, we combined emissions and transport for the assessment of OVOC contributions.

7.         Line 23: I think that the study did not include any statistical tests for significance. It might be interesting to do so if sufficient data points are available. If not, please reword and avoid phrases with "significantly".

Response: Thanks for your valuable suggestion. Here, the terms "significantly" should be removed to avoid misleading. We have revised the sentence, please refer to Lines 23-24:

"Daytime $R_{NetProd}$ is the highest at the urban site (5.9 ppb h$^{-1}$), while nighttime $R_{Emis\&Trans}$ is most negative at the industrial site (0.76 ppb h$^{-1}$)."

8.         Line 24: Typo: overestimates

Response: Thanks for pointing out the error. The related description has been revised, please refer to Lines 24-25:

"Simulations without OVOCs constraint overestimate OVOCs (42.1%~126.5%) and…"

1. Introduction

9.      Line 32: Meaningful digits?

Response: Thank you for pointing out this issue regarding the number of meaningful digits. Considering the uncertainties in the measurements and the proportions, we think it is reasonable to retain one decimal place.

10.     Line 42: My suggestion for a more precise formulation … which leads to a large production of $HO_x$…

Response: Thanks for the suggestion. We agree that a more precise formation would improve the clarity of the sentence. The related description has been revised, please refer to Lines 41-44:

"The study of Li et al. (2021b) indicates that the fast generation of $O_3$ during winter haze in the North China Plain is mainly driven by the photolysis of formaldehyde (HCHO), which leads to a large production of $HO_x$ radical and offsets the radical titration induced by $NO_x$ emissions."

11.     Line 47 (and following) highlight the sources of OVOC. Please, present a complete list here and do not only show examples. Furthermore, the introduction would benefit from integrating the specific OVOC that were highlighted in the studies that you are referring to.

Response: We agree that providing a complete list of OVOC sources and integrating specific examples from the cited studies would enhance the clarity and comprehensiveness of the text. The related description has been revised, please refer to Lines 47-52:

"OVOCs have complex and diverse sources, including primary emissions from anthropogenic, e.g., vehicle exhausts (Gentner et al., 2013; Legreid et al., 2007; Wang et al., 2022b), volatile chemical product use (Ou et al., 2015), industries (Wang et al., 2023), biomass combustion (Gilman et al., 2015;

Karl et al., 2007; Li et al., 2014a; Yokelson et al., 2007), and biogenic sources (Ou et al., 2015; Rieksta et al., 2023). They are also formed secondarily through photochemical reactions (Huang et al., 2020; Song et al., 2024; Xia et al., 2021)."

12.  Line 54-55: As the results pick up on the fate of OVOC and their impact on air quality, I recommend to include a short paragraph about sinks and the ozone formation potential here.

Response: Thanks for your thoughtful suggestion. We agree that it would be beneficial to include a short paragraph discussing the sinks of OVOCs and their role in ozone formation potential. The related description has been revised, please refer to Lines 58-63:

"Due to the high share of OVOCs in VOCs, previous studies have reported that OVOCs could contributed 38%~60% of ozone formation potential (OFP) (Liu et al., 2024; Mo et al., 2022; Wang et al., 2022a, 2024). The loss of OVOCs occurs through photolysis, reactions with oxidants (e.g., OH, $NO_3$, and $O_3$), dilution mixing and deposition (Atkinson, 2000; Atkinson and Arey, 2003). Moreover, air mass transport also can significantly affect the mixing ratio of OVOCs."

13.  Line 75 (and following): Please, be careful in the wording as it is important that your results can be generalized and are useful for the entire community, not only for Zibo inhabitants. I think that this is the case and that you can make it more clear here. E.g. you could present your hypothesis here.

Response: Thank you for your valuable feedback. We agree that the wording should emphasize the broader relevance of our findings beyond Zibo and ensure the results are generalizable and informative for the scientific community. To address this, we have revised the text to highlight the importance of comprehensive OVOCs measurements to constrain numerical models. Please refer to Lines 86-90 in the revised descriptions:

"This study hypothesizes that incorporating observational constraints on OVOCs significantly influences the OBM simulations. To evaluate this, a 5-day field campaign was conducted across five representative sites in Zibo. Concentrations of 74 VOC species, including 29 alkanes, 16 aromatics, 9 alkenes, 18 OVOCs, acetylene and isoprene, are obtained. The contributions of secondary formation,

emissions/transport to OVOCs level are analyzed by the OBM. Additionally, the impact of OVOCs on radical chemistry, atmospheric oxidation capability, and consequently $O_3$ production are quantified."

2. Methodology

14.  Line 99: Here, in Figure 1 and in Table S1 information about the sites is provided. Did you have also wind data from the measurement stations directly? Can you comment on the possibility that site-scale wind conditions influenced the VOC and OVOC observations?

  Response: Thank you for the insightful question. We have conducted statistics on wind conditions, OVOC and VOC levels. We found that site-scale wind patterns have indeed affected the level and spatial variability of OVOCs and VOCs across sites. The relevant descriptions have been added in our manuscript. Please refer to Lines 107-129:

  "Site-scale wind patterns can affect the levels and spatial distribution of OVOCs and PAMS (target VOC species from the Photochemical Assessment Monitoring Stations, including include 29 alkanes, 16 aromatics, 9 alkenes, isoprene, and acetylene) across sites. Urban (ZL) and downwind (CD) sites are impacted by OVOC pollution from northeasterly (NNE, NE, ENE) winds, while the upwind (CQ) site experiences higher OVOC and VOCs pollution under both northeasterly and northwesterly (WNW, NW, NNW) winds (Figure S2 (a, b)). Suburban (TZ) and industrial (XD) sites exhibited higher OVOC and VOCs levels under southeasterly (SE) and southwesterly (WSW) winds, respectively, likely due to upwind emissions from nearby industrial sources. WS between 1 and 2 m s$^{-1}$ were most common (40.4%) during the observation period. At suburban (TZ) and industrial (XD) sites, OVOC and VOCs levels were lower at low wind speeds (WS < 2 m s$^{-1}$) than that at higher wind speeds, reflecting the influence of local emissions (Figure S2(c, d)). In contrast, at downwind (CD) site, higher OVOC and VOCs levels were observed at WS > 2 m s$^{-1}$, indicating the impact of regional transport. At the urban (ZL) site, higher WS are associated with lower VOCs levels and higher OVOC levels, indicating the influence of aging air masses transported from upwind regions."

[Figure]

Figure S2 Concentration statistics of OVOC (a, c) and PAMS (b, d) at different wind speed (WS) and wind direction (WD) intervals, respectively.

Line 107 (and following): Please explain PAMS. I prefer to have a full list of VOC and OVOC in the SI, including information about calibration, response factors, extraction efficiency (for the OVOC) and uncertainties. Please, provide details about the sampling line. Did you remove ozone to avoid losses of the VOC or OVOC?

Response: We sincerely appreciate your insightful comments and suggestions. We agree to clarify the term "PAMS" and provide detailed information about VOC and OVOC measurements. A full list of VOCs and OVOC has been added in the supporting information (Table S2) (showed in general comments 1). The related descriptions on VOCs and OVOC measurements methods have been added, please refer to Lines 129-146:

"A mixture of 56 PAMS target species (Spectra Gases Inc., USA, Table S2) was used for the calibration of the GC-FID system. Each VOC analyzer provided measurements with a 1-hour temporal resolution. More detailed descriptions of these instruments can be found in previous studies (Li et al., 2023; Wang et al., 2014; Yang et al., 2022; Zheng et al., 2023). Overall, the detection limits for most VOC species are below 0.1 ppb. Additionally, 18 oxygenated VOCs species were collected by 2,4-dinitrophenylhydrazine (DNPH) sorbent tubes in conjunction with an automated sampler for a period of 1 or 3 hour per sample. A 47 mm quartz filter membrane is attached to the front of the sampling tube to filter particulate matter. An ozone scrubber (silica gel column tubing coated with potassium iodide) was placed at the front of the air inlet to avoid ozone interference. OVOCs were derivatized in cartridges to hydrazones during sampling. The cartridges were eluted with 3 mL of acetonitrile and stored at 0-4 °C immediately. Then the eluants were analyzed using an Agilent HPLC, equipped with ultraviolet absorption detector (UVD), quadruple pump, and Agilent TM C18 reversed column (250 mm×4.6 mm, 5.0 µm). A gradient elution was used, and the mobile phase was mixing of acetonitrile, tetrahydrofuran and water. The analysis was carried out using a ternary gradient elution program at a flow rate of 1.2 mL/min, with detection wavelength of 360 nm, and sample volume of 10 µL at a column temperature of 45 °C. More details about OVOC samplings and analysis can be found in Peng et al. (2023). The lower limit of detection for OVOCs were <0.1 ppb (Peng et al., 2023)."

Regarding ozone removal, there is an ozone scrubber (silica gel column tubing coated with potassium iodide) at the air inlet to avoid ozone interference during OVOC sampling. While the on-line VOC analysis system does not remove $O_3$, we believe that the impact of ozone on the measurement of volatile organic compounds (VOCs) at an hourly sampling frequency is negligible.

15. Line 127: Typo: Coupled

Response: Thanks for pointing out the errors. The related description has been revised, please refer to Line 161:

"A box model (F0AM) coupled with the Master Chemical Mechanism (MCM) v3.3.1 was …"

16.  Line 154 (and following): The variation of atmospheric mixing ratios of OVOC depends on its sources and sinks. The list here includes photochemical production and emissions (both sources), regional transport (both source and sink), and deposition (sink). Why did you combine emissions and transport in this budget? Why is the photolysis of OVOC not included as a loss? I checked the two references provided here and they did not help me understanding the approach. Tan et al. (2018a) used the same OBM, but for assessing ozone production and loss. They refer to Tan et al. (2017) who have used the model again in a different context and highlighted large uncertainties of the model calculations. Xue et al. (2014) used the OBM as well and described it in detail in their SI. However, again, I cannot find information about OVOC or the budget approach analysing production and loss terms. I find this method description crucial for understanding the results parts (3.2). With the currently provided information and description, I do not fully understand the approach and cannot judge the quality of the results.

Equations (1) and (2): In line with my previous remark, the reader needs more information and a detailed explanation/justification about the choice of production and loss terms. Please provide a suitable reference, in addition.

Response: We sincerely apologize for the confusion caused by our misrepresentation of the OVOC budget approach and the lack of detailed explanations in the manuscript. The chemical processes of OVOCs include chemical production ($R_{ChemProd}$) by VOC oxidation (source), and chemical loss ($R_{ChemLoss}$), such as OVOC photolysis (sink) and reactions with oxidants (OH, $NO_3$ and $O_3$) (sink) (https://mcm.york.ac.uk/MCM/, last access: 13 Jan 2025) (Atkinson, 2000; Atkinson and Arey, 2003; Jenkin et al., 2015; Saunders et al., 2003). Those chemical processes are combined into a net chemical production ($R_{NetProd} = R_{ChemProd} - R_{ChemLoss}$). Photolysis is indeed a critical loss pathway for OVOCs and was accounted for in our calculations as a part of $R_{NetProd}$. We regret not explicitly mentioning this in the manuscript and have revised the related text to ensure clarity.

For the combination of $R_{Emis}$ and $R_{Trans}$, as we response for major comment 2, OVOCs can be directly emitted from surface source ($R_{Emis}$), which is different from $O_3$. According to the method of Xue et al., 2014b, the sum of $R_{Trans}$ and $R_{Emis}$ should equal to $R_{meas}$ minus Rchem and $R_{dep}$. However,

the 0-D observation-based model has a limitation of not considering transport and emissions. Therefore, the emissions and transport are combined in our calculations. The related descriptions have been revised, please refer to Lines 200-218:

"At a given site, variations in OVOCs mixing ratios are mainly influenced by in-situ photochemical production and chemical loss, emissions, regional transport, and deposition (Tan et al., 2018a; Xue et al., 2014a; Zhang et al., 2021). The change rate of observed OVOCs ($R_{Meas}$) is calculated by Equation (1). The in-situ photochemical production of OVOC ($R_{ChemProd}$) is mainly caused by the oxidation of VOCs, while their in-situ chemical loss ($R_{ChemLoss}$) includes photolysis and reactions with oxidants (OH, $NO_3$, and $O_3$) (https://mcm.york.ac.uk/MCM/, last access: 13 Jan 2025) (Atkinson, 2000; Atkinson and Arey, 2003; Jenkin et al., 2015; Saunders et al., 2003). The in-situ net OVOCs chemical production ($R_{NetProd}$) (Equation (2)) and their removal by deposition ($R_{Dep}$) are calculated hourly according to the OBM simulation. The OBM primarily accounts for atmospheric photochemical reactions, and deposition within the boundary layer. However, previous studies have reported that the OBM lacks an explicit representation of transport processes and emissions (Wolfe et al., 2016; Zhang et al., 2021), making it challenging to disentangle their respective contributions. Therefore, emissions and transport are combined to a single term ($R_{Emis\&Trans}$) to represent their contributions collectively. If the $R_{Emis\&Trans}$ is positive, it is considered a net import of emissions/transport, whereas a negative suggests a net export. The emissions and regional transport of OVOCs ($R_{Emis\&Trans}$) are computed as Equation (3).

$$R_{Meas} = \sum_i \frac{d([OVOC]_i)}{dt} \tag{1}$$

$$R_{NetProd} = \sum_i (R_{ChemProd,i} - R_{ChemLoss,i}) \tag{2}$$

$$R_{Emis\&Trans} = (R_{Meas} - R_{NetProd} - \sum_i R_{Dep,i}) \tag{3}$$

where $[OVOC]_i$ is the mixing ratios of OVOC species $i$ constrained in OBM, 15 in total (Table S2). dt is the time-step of modeling, $d[OVOC]_i$ refer to the change in mixing ratio of OVOC species $i$."

3. Results and discussion

17.  Line 169 (and thereafter): You are reporting mixing ratios of VOC and OVOC in ppbV. Do not change the wording to concentrations.

Response: Thank you for highlighting this key point. We acknowledge that "mixing ratios" is the correct term when reporting VOC and OVOC values in ppbV. We have revised the wording accordingly in the manuscript. Please refer to Line 259:

"The mean VOCs mixing ratio in this study is 44.6±20.9 ppb…"

Additionally, all instances of inaccurate usage of "concentration" have been corrected to avoid confusion. Thank you for your careful review.

18.  Line 169 (and thereafter): What is the uncertainty of the VOC and OVOC measurement? Is it meaningful to report the first digit after the decimal point?

Response: The minimum detection limits for VOCs are <0.1 ppb, and we think it makes sense to keep one decimal place.

19.  Line 173: What is the upper middle range? And why do you conclude that this indicates strong anthropogenic VOC emissions at Zibo city?

Response: We are sorry for the mistake in our description. The "upper middle range" refers to the higher VOCs mixing ratios observed in Zibo compared to the median (~32 ppb) reported from other cities in China, as shown in Table S4 and Figure 3. In addition, industrial sectors contribute approximately half of its annual Gross Domestic Product (GDP) in Zibo (Li et al., 2017; Ren, 2011), indicating the strong VOC levels in the city are strongly influenced by anthropogenic emissions.
The related descriptions have been revised, please refer to Lines 262-275:

"Compared with the median VOC levels (~32 ppb) in other cities in China (Figure 1, Table S4), VOC levels in Zibo is in the upper-middle range. Previous studies have demonstrated that industrial processes account for approximately 49% of total VOC emissions in Shandong Province (Jiang et al., 2020; Li et al., 2017; Ren, 2011; Zheng et al., 2021). This indicates strong anthropogenic VOCs emission in Zibo. Notably, VOC emission intensity in Zibo was among the highest in Shandong

Province, with values > 90 t km$^{-2}$ y$^{-1}$, even >108 t km$^{-2}$ y$^{-1}$ in some areas in 2016 (Jiang et al., 2020; Zhou et al., 2021)."

[Figure]

**Figure 1. Comparison of VOC mixing ratios and compositions in this study with former studies based on Table S4. The red dash line represents the median levels (~32 ppb) of VOCs。**

20.  Line 177: Daytime maximum temperature is reported here the second time (compare line 163).

Response: Thank you for pointing out the repetition. We have removed the sentence of line 177 and consolidated the relevant details into line 257 for clarity and conciseness.

The related descriptions have been revised, please refer to Lines 251-253 in the update manuscript:

"The field campaign is characterized by consistent hot and sunny conditions, with the average daily maximum temperature and SSR of 32.2±1.4 °C (peak at 34.1 °C) and 2.1±0.4×10$^6$ J m$^{-2}$ (**Error! Reference source not found.**, Figure S2 (a)), respectively, which favors the photochemical formation of O$_3$."

21.     Line 179: NO$_2$ mixing ratios

Response: Thanks for the comment. We have revised the description. The related descriptions have been revised, please refer to Lines 271-272:

"In addition, the difference between peak and valley NO$_2$ mixing ratios was 14.4±3.2 ppb, indicating that substantial NO$_x$ was converted to O$_3$."

22.     Line 183: What do you mean with "the timing of the PAMS data was matched to that of the OVOCs data"?

Response: We are sorry for the lack of precise descriptions. Since VOC observations were conducted at a 1-hour resolution, while OVOC offline measurements were typically taken every 3 hours per sample. In order to better assess TVOC levels and ensure comparability between VOC and OVOC data, VOC data corresponding to the observation times of OVOCs were extracted. "the timing of the PAMS data was matched to that of the OVOCs data," means that the hourly VOC data was averaged to match the time of OVOCs data, which were sampled every 1 or 3 hours. This matching ensured consistency in temporal resolution, allowing for direct comparisons and a more accurate assessment of the relationship between VOCs and OVOCs. The related descriptions have been revised, please refer to Lines 276-278:

"The hourly PAMS (including alkanes, alkenes, aromatics, acetylene, and isoprene) data were aligned with the 1/3-hour sampling intervals of the OVOCs data to ensure comparability between the two datasets."

23.     Line 188 (and similar starts of sentences): Please check for correct English formulation. "As for individual side" does not sound correct.

Response: Thanks for pointing out the shortcomings. We have checked the manuscript for similar beginnings and revised it. For the revise made here, please refer to Line 293:

"Across the five sites, the average O$_3$ mixing ratios are comparable across all the sites…"

24.    Line 188: O$_3$ mixing ratios

Response: Thanks for pointing out the error. This and other references to concentration have been revised.

25.    Lines 189 – 212: The paragraph describes the results very qualitatively with relative wording as "slightly lower", "comparable levels", "higher than". Is any of the observed differences significant? How many data points to you have? Can you do a statistical analysis with a significance test? Can you focus this paragraph on the main message and discussion point?

Response: Thank you for your helpful suggestion. We have revised the paragraph to incorporate statistical analysis results. We performed one-way ANOVA and Tukey's HSD post hoc tests to evaluate the significance of differences in pollutant concentrations across the sites. Our results showed that while O$_3$ concentrations did not significantly differ between sites (p=0.366), significant differences were observed for NO$_2$, NO, CO, and TVOCs. The description of time series varies is based only on instantaneous data and no significance analysis is performed. The relevant descriptions have been revised for main message. Please refer to Lines 286-306:

"Across the five sites, the O$_3$ mixing ratios are comparable across all the sites (Table S6, Table S11) (p > 0.05). However, TVOCs at suburban site (TZ, 58.5±35.0 ppb) is the highest (Figure (c), Figure S4 (b)), which is attributed to oil refineries near this site. The downwind site (CD) has slightly lower NO$_2$ level (10.8±5.1 ppb) and lower TVOCs mixing ratios (35.7±12.5 ppb) than urban site (ZL, 14.8±6.5 and 40.6±10.3 ppb, respectively) and upwind site (CQ, 12.7±8.1 and 42.3±15.4 ppb, respectively), and has higher O$_3$ mixing ratio (58.6±30.0 ppb) than CQ and ZL station. This may be attributed to the sequential transport of O$_3$ and its precursors from the upwind station (CQ) to urban station (ZL), and subsequently to the downwind station (CQ), driven by the dominant northeasterly winds (**Error! Reference source not found.** (b), Figure S1).

According to the time series of individual pollutant (Figure S3 (b)), CQ showed obvious peak mixing ratios of O$_3$, NO$_2$, NO and CO than the other sites during August 8-9, with stagnant conditions

(WS < 2 m s$^{-1}$), indicating stronger emissions from combustion sources and possibly fast photochemical process near CQ. In addition, XD showed high mixing ratios of CO during August 8-9, and high daytime TVOCs levels on August 9 (9:00-14:00 LT, 90~110 ppb). Given CO's relatively inert nature and the absence of similar CO peaks at the other four sites, the abnormal CO peak at XD is related to strong emissions from nearby factories in the industrial park. TZ showed distinct morning and evening peaks of TVOCs at 6:00 LT (163.0 ppb) and 21:00 LT (120.0 ppb) on August 8, and a night peak at 1:00 LT on August 10 (130.3 ppb), which were attributed to emissions from the neighboring oil field operations. From August 10 to 12, as wind speeds increased, pollutants levels at all sites decreased to similar levels. Overall, local anthropogenic emissions in Zibo were more prominent under weak wind conditions."

**Table S2 The one-way analysis of variance (ANOVA) results for pollutants level, OFP of different VOC groups, and daytime contributions of OVOC budget.**

| Type | Group | F-stat[a] | p-value[b] |
|---|---|---|---|
| Mixing ratio | O$_3$ | 1.08 | 3.66E-01 |
| | NO$_2$ | 8.52 | 1.09E-06 |
| | NO | 10.74 | 2.07E-08 |
| | CO | 33.52 | 3.09E-25 |
| | TVOCs | 8.78 | 1.35E-06 |
| OFP | OVOCs | 8.28 | 3.06E-06 |
| | Aromatics | 28.56 | 4.06E-19 |
| | Alkanes | 28.23 | 6.21E-19 |
| | Alkenes | 15.48 | 3.55E-11 |
| | BVOCs | 13.11 | 1.36E-09 |
| | Alkyne | 39.91 | 4.09E-25 |
| | TVOCs | 8.10 | 4.15E-06 |
| Daytime OVOC budget | R$_{NetProd}$ | 14.87 | 5.42E-11 |
| | R$_{Dep}$ | 4.13 | 2.89E-03 |
| | R$_{Emis\&Trans}$ | 10.74 | 4.35E-08 |

Note: [a] F-statistic (F-stat) measures the ratio of OFP variance between VOC groups to the variance within the groups. A higher F-stat indicates a larger difference between the groups relative to the variation within the groups. This suggests that OFP is more likely to differ across VOCs categories.
[b] p-value indicates the probability that the observed difference (or a more extreme difference) occurred by chance. A p-value less than 0.05 typically indicates that the observed differences are statistically significant, meaning there is a high likelihood that the differences are not due to random variation.

26.  Lines 213—220: The mean OFP in Zibo were 410 µg m$^{-3}$ with CQ (464 µg m$^{-3}$), TZ (456 µg m$^{-3}$), ZL (441 µg m$^{-3}$), XD (422 µg m$^{-3}$) and CD (279 µg m$^{-3}$). So all stations were similar except of CD, correct? Did you do a significance test? In the conclusions (lines 345-347) other numbers are stated. What is correct?

Response: Thank you for your conductive question and detailed review. For question 1 and 2, We have added a significance test. The OFP contributed by TVOCs were similar for all stations except CD. However, there were differences in the OFP contributed by VOC subgroups. For the differences in the OFP contributions of VOC and its subclasses at each station, we have conducted a significant test and modified the corresponding expressions. Please refer to Lines 307-332:

"To compare the secondary O$_3$ formation in each site, the ozone formation potential (OFP) of each VOCs is calculated (Equation (6)). The mean OFP in Zibo during the observation is 410.4±197.2 µg m$^{-3}$, with OVOCs accounting for the largest proportion (31.5%~55.9%), followed by aromatics (10.2%~41.2%). Alkanes (10.3%–24.6%) and alkenes (11.4%~23.1%) make comparable proportions, while BVOCs accounted for only 2.1%~7.6% of the total OFP (Figure 5 (b)). The one-way analysis of variance (ANOVA) results (p < 0.05) indicate significant differences (Armstrong et al., 2000) in VOC subclass contribution to OFP across the 5 sites. Alkanes and aromatics show larger F-values (Table S11), reflecting greater variations in the contributions to OFP across the 5 sites, whereas OVOC and BVOC (isoprene) exhibited lower variability. Post-hoc Tukey honestly significant difference (HSD) tests were performed followed ANOVA to further identify specific significant differences in VOC subcategories between sites (Figure S14). The OFP of TVOCs is generally similar across stations, except for the downwind station (CD). The upwind station (CQ, 464.2±162.3 µg m$^{-3}$) has the highest OFP, followed by the suburban site (TZ, 456.3±295.3 µg m$^{-3}$), the urban site (ZL, 441.1±174.5 µg m$^{-3}$), the industrial site (XD, 422.9±166.9 µg m$^{-3}$), and the downwind site (CD, 279.4±101.2 µg m$^{-3}$) (Table S5). Differences in OFP levels of aromatics and alkanes at downwind station (CD), suburban station (TZ) and industrial station (XD) are minimal (Figure S14 (b, c)). However, significant differences in OFP contributed by OVOC at downwind station (CD) compared to suburban station (TZ) and industrial station (XD) are attributed to OVOC emission sources, regional transport and secondary formation (Figure S14 (a)).

Apart from CQ, OVOCs are the dominant contributors to OFP at each site, especially TZ and XD, with mean OFP of 254.9±276.1 µg m$^{-3}$ (55.9%) and 194.7±101.0 µg m$^{-3}$ (46.0%) from OVOCs, respectively. This indicates the key role of OVOCs in the formation of O$_3$ at our observational sites. Among OVOC species, HCHO is the dominant contributor to OFP across the five sites (56.6~202.0 µg m$^{-3}$). This is consistent with previous studies (Duan et al., 2008; Huang et al., 2020; Zhou et al., 2024). The top four OVOC species are formaldehyde, acetaldehyde, propionaldehyde, and butyraldehyde, which cumulatively contributed 91%~95% of the OFP from OVOCs (Table S5)."

[Figure]

**Figure S14 Comparison of Tukey honestly significant difference (HSD) tests for OFP of VOC and its subclasses between different sites. Blue Dots represent the mean difference between the two sites, blue error bar represents the 95th percentile confidence interval (CI), red dots indicate significant difference between the two sites.**

**Table S11 The one-way analysis of variance (ANOVA) results for pollutants mixing ratios, OFP of different VOC groups, and daytime contributions of OVOC budget.**

| Type | Group | F-stat[a] | p-value[b] |
|---|---|---|---|
| Mixing ratio | O$_3$ | 1.08 | 3.66E-01 |

|  |  | F-stat[a] | p-value[b] |
| --- | --- | --- | --- |
|  | $NO_2$ | 8.52 | 1.09E-06 |
|  | NO | 10.74 | 2.07E-08 |
|  | CO | 33.52 | 3.09E-25 |
|  | TVOCs | 8.78 | 1.35E-06 |
| OFP | OVOCs | 8.28 | 3.06E-06 |
|  | Aromatics | 28.56 | 4.06E-19 |
|  | Alkanes | 28.23 | 6.21E-19 |
|  | Alkenes | 15.48 | 3.55E-11 |
|  | BVOCs | 13.11 | 1.36E-09 |
|  | Alkyne | 39.91 | 4.09E-25 |
|  | TVOCs | 8.10 | 4.15E-06 |
| Daytime OVOC budget | $R_{NetProd}$ | 14.87 | 5.42E-11 |
|  | $R_{Dep}$ | 4.13 | 2.89E-03 |
|  | $R_{Emis\&Trans}$ | 10.74 | 4.35E-08 |

Note: [a] F-statistic (F-stat) measures the ratio of OFP variance between VOC groups to the variance within the groups. A higher F-stat indicates a larger difference between the groups relative to the variation within the groups. This suggests that OFP is more likely to differ across VOCs categories. [b] p-value indicates the probability that the observed difference (or a more extreme difference) occurred by chance. A p-value less than 0.05 typically indicates that the observed differences are statistically significant, meaning there is a high likelihood that the differences are not due to random variation.

For question 3, we have checked the numerical descriptions in this section with those in the conclusions. The corresponding numbers in the conclusion have been corrected, please refer to Lines 505-511:

"The OFP in Zibo is 410.4±197.2 µg m$^{-3}$, with OVOCs accounting for the largest proportion (31.5%~55.9%). The upwind site (CQ, 464.2±162.3 µg m$^{-3}$) has the highest OFP, followed by the suburban site (TZ, 456.3±295.3 µg m$^{-3}$), the urban site (ZL, 441.1±174.5 µg m$^{-3}$), the industrial site (XD, 422.9±166.9 µg m$^{-3}$), and the downwind site (CD, 279.4±101.2 µg m$^{-3}$) (Table S5). OFP contributed by OVOCs is most dominant at suburban (TZ, 254.9 µg m$^{-3}$) and industrial (XD, 194.7 µg m$^{-3}$) sites, followed by urban (ZL, 148.9 µg m$^{-3}$), upwind (CQ, 146.2 µg m$^{-3}$), and downwind (CD, 102.3 µg m$^{-3}$) sites."

27.     Line 220: Typo: key role.

Response: Thanks for pointing out the mistake. The spelling has been revised, please refer to Line 328:

"This indicates the key role of OVOCs in the formation of O$_3$ at our observational sites."

28.    Figure 4: The two panels show the ozone formation potential attributed to measured VOC and OVOC in absolute and relative proportions. They do not show the concentrations. However, it would be interesting to have the same stacked bar plot for the mixing ratios in addition (e.g. as a third panel).

Response: Thanks for pointing this out. We are sorry for the misrepresentation of OFP's level. You are correct that the current panels focus on the ozone formation potential (OFP) in absolute and relative terms, rather than concentrations. We have incorporated the stacked bar plot for VOC groups' mixing ratios, please refer to Figure 5.

[Figure]

**Figure 5. (a) Ozone formation potential (OFP) and (b) proportions of OFP contributed by VOC subgroups, along with (c) mixing ratios of VOC subgroups at five sites.**

29.    Line 237: What do you want to say with this first sentence? Please rephrase.

Response: We are sorry that the current description of the first sentence in Line 237 is somewhat unclear. We have revised this sentence for clarity. Please refer to Lines 347-348:

"OBM simulation results were used to analyze the contributions of chemical processes, and emissions/transport to OVOCs."

30. Line 238: What do you mean with saying that "all five sites show good model performance"?

Response: We apologize for the unclear description. The modeled $O_3$ in the Base scenario shows good agreement with the observations at the five sites, which demonstrates good model performance. The revised descriptions have been revised, please refer to Lines 348-351:

"Overall, the modeled $O_3$ in the Base scenario exhibited good model performance at the five sites, with R values exceeding 0.85 and IOA values greater than 0.80 (Table S7). These metrics indicate a high level of agreement between observed and modeled data, comparable to results reported in previous studies (Qin et al., 2023; Zheng et al., 2023)"

31. Lines 238: What "R values"? What "IOA values"? I cannot see those in Figure S5.

Response: Thank you for pointing out this discrepancy. You are correct that the reference to "Figure S5" is a mistake. It should have referred to "Table S7." We apologize for this error, and the manuscript has been corrected accordingly. Please refer to Lines 3448-349:

"Overall, the modeled $O_3$ in the Base scenario exhibited good model performance at the five sites, with R values exceeding 0.85 and IOA values greater than 0.80 (Table S7)."

**Table S7 Modeled $O_3$ assessment of Base and Free scenario.**

| Site | Base | | Free | |
|------|------|------|------|------|
|      | IOA | R | IOA | R |
| CD | 0.80 | 0.88 | 0.90 | 0.88 |
| CQ | 0.90 | 0.87 | 0.86 | 0.87 |
| TZ | 0.88 | 0.88 | 0.85 | 0.88 |
| XD | 0.86 | 0.88 | 0.83 | 0.89 |
| ZL | 0.88 | 0.89 | 0.88 | 0.87 |

32.    Line 243: This sentence needs to be phrased very precisely, as I think it could be misleading otherwise.

Response: We are sorry for the misleading words. We have added the methodology for the OVOC source and sink calculations in Section 2.3. Please refer to Lines 201-218:

"At a given site, variations in OVOCs mixing ratios are mainly influenced by in-situ photochemical production and chemical loss, emissions, regional transport, and deposition (Tan et al., 2018a; Xue et al., 2014a; Zhang et al., 2021). The change rate of observed OVOCs ($R_{Meas}$) is calculated by Equation (1). The in-situ photochemical production of OVOC ($R_{ChemProd}$) is mainly caused by the oxidation of VOCs, while their in-situ chemical loss ($R_{ChemLoss}$) includes photolysis and reactions with oxidants (OH, $NO_3$, and $O_3$) (https://mcm.york.ac.uk/MCM/, last access: 13 Jan 2025) (Atkinson, 2000; Atkinson and Arey, 2003; Jenkin et al., 2015; Saunders et al., 2003). The in-situ net OVOCs chemical production ($R_{NetProd}$) (Equation (2)) and their removal by deposition ($R_{Dep}$) are calculated hourly according to the OBM simulation. The OBM primarily accounts for atmospheric photochemical reactions, and deposition within the boundary layer. However, previous studies have reported that the OBM lacks an explicit representation of transport processes and emissions (Wolfe et al., 2016; Zhang et al., 2021), making it challenging to disentangle their respective contributions. Therefore, emissions and transport are combined to a single term ($R_{Emis\&Trans}$) to represent their contributions collectively. If the $R_{Emis\&Trans}$ is positive, it is considered a net import of emissions/transport, whereas a negative suggests a net export. The emissions and regional transport of OVOCs ($R_{Emis\&Trans}$) are computed as Equation (3).

$$R_{Meas} = \sum_i \frac{d([OVOC]_i)}{dt} \tag{1}$$

$$R_{NetProd} = \sum_i (R_{ChemProd,i} - R_{ChemLoss,i}) \tag{2}$$

$$R_{Emis\&Trans} = (R_{Meas} - R_{NetProd} - \sum_i R_{Dep,i}) \tag{3}$$

where $[OVOC]_i$ is the mixing ratios of OVOC species $i$ constrained in OBM, 15 in total (Table S2). dt is the time-step of modeling, $d[OVOC]_i$ refer to the change in mixing ratio of OVOC species $i$."

33.  Line 245/246: What do you mean with "mainly concentrated"?

Response: We apologize for the unclear presentation. The "mainly concentrated" intends to convey that the contributions of $R_{NetProd}$ predominantly occur during the daytime. The relevant descriptions have been revised, please refer to Lines 251-252:

"The contributions of $R_{NetProd}$ predominantly occur during the daytime (Figure 6)."

34.  Lines 242-263: This discussion is difficult to follow overall as it is rather qualitative and I did not understand the methodology in the first place. In general, can you elaborate on the sensitivity of dOVOC/dt and its sensitivity towards the time-step. Did you use the measurement intervals or hourly data from the model?

Response: We apologize for the lack of clarity in the description of OVOC budget, we have revised the methodology in detail in Lines 193-211 of the manuscript. The data used for OVOC budget analysis are hourly data obtained from OBM output based on a 1-hour time step. Since OVOC budget calculations involve time-step, which could have an impact on the results of OVOC budget. Therefore, we have tested the sensitivity of the simulated time-step based on one of the sites (urban site, ZL). We have added simulations with time steps of 30, 10 and 5minutes, corresponding to OVOC budget calculations using simulated resolution data. A description has been added to the uncertainty analysis in Section 3.4, see Lines 495-505:

"When analyzing the contribution of chemical pathways, emission/transport and deposition to OVOCs according to Equations (1-3), the OVOC budget may be affected by the modeling time step. A sensitivity analysis was conducted at ZL, with simulations at different time steps (30 min, 10 min , and 5 min). As shown in Figure S13, the diurnal trends of the chemical contributions ($R_{NetProd}$), emission/transport ($R_{Emis\&Trans}$) and deposition ($R_{Dep}$) to OVOC are similar. The magnitude of the instantaneous change in $R_{Emis\&Trans}$ decreases when the time step is shortened. Specifically, the contributions of $R_{NetProd}$ and $R_{Emis\&Trans}$ to OVOC increased, while the contribution of $R_{Dep}$ decreased with shorter time steps (Table S10). When the time step was reduced to 5 minutes, the contributions of

$R_{NetProd}$ and $R_{Emis\&Trans}$ to OVOC increased by 4.3% and 5.0%, respectively, while the contribution of $R_{Dep}$ decreased by 0.2%. Therefore, shortening the time step in the model simulation may result in limited increase contribution from $R_{NetProd}$ and $R_{Emis\&Trans}$ to OVOC."

[Figure]

**Figure S13 Comparison of $R_{NetProd}$, $R_{Emis\&Trans}$, and $R_{Dep}$ contributed to OVOCs for different time-step.**

**Table S10 Changes in $R_{NetProd}$, $R_{Emis\&Trans}$, and $R_{Dep}$ contributions to OVOC in different time-step scenarios relative to the Base scenario (1-hour time-step).**

| Time-step | $R_{NetProd}$ | $R_{Emisa\&Trans}$ | $R_{Dep}$ |
|---|---|---|---|
| 5 minutes | 4.3% | 5.0% | -0.2% |
| 10 minutes | 4.2% | 4.8% | -0.1% |
| 30 minutes | 2.8% | 3.2% | -0.1% |

35.    Line 272: Typo:..will result in…

Response: Thank you for pointing out the typo. The mistake has been corrected. Please refer to Lines 376-379:

"It has been shown that the box model, which did not take into account transport (including horizontal and vertical diffusion) and emissions, will result in overestimations of OVOCs, peroxyl radical and PAN (Qu et al., 2021)."

36.    Line 274: Are the fractions given here the comparison of the Free Scenario with the Base case?

Response: Yes, the fractions given here are indeed a comparison of the Free Scenario with the Base case. To improve clarity, we have revised this sentence. Please refer to Lines 379-381:

"In this study, OVOCs are overestimated by 42.1%~126.5% in the Free scenario compared with the Base scenario (Figure 6 (a), Figure S6 (c)), especially HCHO (76.3%) and benzaldehyde (737.5%)."

37.     Line 277: How large was the fraction of HCHO of the total OVOC measured in general?

Response: The measured HCHO at the five sites in this study account for 41.7%~72.0% of total OVOCs. Previous studies have shown HCHO to be 29%~50% of OVOC (Rao et al., 2016; Shen et al., 2021).

38.     Line 280/281: I don't understand this sentence. Do you mean the chemical reactions that include $RO_2$, $HO_2$ and OH, which are known and/or implemented in the MCM mechanism? This is certainly not the same.

Response: We apologize for the misrepresentation. Yes, I mean the chemical reactions that include $RO_2$, $HO_2$ and OH, which are implemented in the MCM mechanism. Please refer to Lines 386-389:

"To assess the impact of OVOCs on the simulation of $RO_x$ species ($RO_2$, $HO_2$, and OH), the chemical budgets of these species, as influenced by OVOCs, are quantified according to Liu et al. (2012) and Xue et al. (2016) (**Error! Reference source not found.** (b), Figure S8 (a))."

39.     Lines 282-306: Please, be precise in the presentation of results and avoid relative wording. It is difficult to follow here the discussion and to find out what the key message is.

Response: We thank you for pointing out the need for greater precision and clarity in presenting the results. In the original manuscript of Lines 282-306, we intended to illustrate the OVOC photolysis pathway to promote the generation of peroxide radicals ($RO_2$ and $HO_2$) through the RO$x$ budget, and the combined effect of OVOC photolysis on OH with two different effects. We have revised the relevant descriptions to make it precise, please refer to Lines 390-411 of the revised manuscript.

"In the Free scenario, the daytime net production of $RO_x$ ($P(RO_x)$) was estimated to range from $0.03$~$0.14$ ppb h$^{-1}$ across four sites (excluding TZ), indicating an overestimation of $RO_x$. Notably, the TZ site exhibited negative P(ROx) values, suggesting the potential existence of unaccounted $RO_x$ sources in this region. The mean daytime $P(RO_x)$ in the Free scenario was calculated as $4.8\pm2.7$ ppb h$^{-1}$, 18.8% higher than that in the Base scenario ($4.0\pm2.3$ ppb h$^{-1}$). As illustrated in Figure 8 and Figure

S8(a), the photolysis of OVOCs (including HCHO) dominants $P(RO_x)$, with a mean rate of $2.9\pm1.9$ ppb $h^{-1}$ in the Free scenario, 27.4% higher than that in Base scenario ($2.3\pm1.5$ ppb $h^{-1}$). This substantial increase in OVOCs photolysis consequently amplified the formation of peroxyl radicals ($RO_2$ and $HO_2$). Among the production pathways, the photolysis of HCHO demonstrated the most pronounced impact on $HO_2$ production in the Free scenario ($0.1\sim1.9$ ppb $h^{-1}$), with an increase of $7.8\%\sim151.2\%$ ($0.1\sim1.2$ ppb $h^{-1}$) than in the Base scenario ($0.5\sim1.1$ ppb $h^{-1}$) (**Error! Reference source not found.** (b)).

The interference of OVOCs on OH is comprehensive. On the one hand, increased OVOCs tends to elevate the generation of $HO_2$, which can directly or indirectly boost OH generation via the reaction of NO (Figure S9). On the other hand, the higher OVOCs levels can decrease OH via the reaction of OH+OVOCs (Qu et al., 2021; Tan et al., 2019b). In the Free scenario, total OH sources (including $H_2O_2$+hv, HONO+hv, $O_3$+hv, and $HO_2$+NO) is $7.5\sim12.1$ ppb $h^{-1}$, which is $0.3\sim1.1$ ppb $h^{-1}$ higher than that in the Base scenario ($10.2\sim11.0$ ppb $h^{-1}$) (Figure S9). Conversely, OH destruction to peroxyl radicals in the Free scenario ($7.1\sim11.8$ ppb $h^{-1}$) is $0.3\sim2.1$ ppb $h^{-1}$ higher than that in Base scenario ($6.1\sim9.7$ ppb $h^{-1}$), leading to a net OH loss of $0.1\sim0.9$ ppb $h^{-1}$. This underestimation of OH without OVOCs constraint biases atmospheric oxidation capacity (AOC) by $0.1\%\sim10.0\%$ (excluding XD) (Figure S10), affecting the evaluation of VOCs decay via OH oxidation (Li et al., 2022)."

40.     Figure 7, Figure 8 and discussion in the text: These are certainly interesting results, visualizing the different production and loss pathways both for $RO_x$ and $O_3$ production. How much do uncertainties in the model and the chemical mechanism play a role for the interpretation? What are the limitations of your approach for (1) the site comparison and for (2) the role of OVOC in the air quality analysis?

Response: Thank you for the insightful comments regarding the uncertainties in the model and chemical mechanism. Below, we address these points in detail.

For question 1, we have conducted sensitivity tests in Section 3.4 to quantify some of these uncertainties, focusing on $NO_x$ settings, HONO values, and time-step variations. These tests revealed that uncertainties in $NO_x$ settings could lead to variations of $-12\%\sim7\%$ in daytime $O_3$ and $-30\%\sim10\%$ in the net chemical production rate of OVOCs. Uncertainties in $HONO/NO_2$ ratios resulted in deviations

of up to 5.4%, 5.6%, 7.2% and 3.8% for daytime OH, HO$_2$, RO$_2$, and O$_3$ levels, respectively. In addition, a sensitivity test for time-step variations showed that shortening the time-step could increase the contributions of R$_{NetProd}$ and R$_{Emis\&Trans}$ to OVOCs by up to 4.3% and 5.0%, respectively. However, uncertainties for the emissions and transport, and incomplete chemistry needs further studies conducted by CTMs and refining chemical mechanisms in OBM, respectively.

For question 2, (1) the limitations for site comparison include spatial heterogeneity, meteorological variability, and the absence of expression of emissions (discussed above). (2) the limitation for the role of OVOC in air quality analysis include measurement uncertainty (OVOC and VOC), and absence of representation of emissions and transport (discussed above).

Above all, future work will require improving the model by refining chemical mechanisms and enhancing the accuracy of observations of VOCs and NO$_x$ in OBM. Regarding the contributions of emissions and transport, future studies can assess these using CTMs.

4. Conclusions

41.     Line 339: What exactly is the upper-middle range of VOC mixing ratios in which cities?

Response: We apologize for the ambiguity regarding the "upper-middle range" of VOC mixing ratios. To clarify, the upper-middle range of VOC mixing ratios refers to levels above the median values (~32 ppb) derived from previous studies (Figure 3, Table S4). These studies encompass various regions in most Chinese cities, offering a statistical basis for comparison.
The relevant description has been revised, please refer to Lines 501-503:

"Compared with previous studies conducted in most Chinese cities, the VOCs level in Zibo is in the upper-middle range (>32 ppb), with OVOCs being the second-largest contributor (29.4%~36.1%) after alkanes (34.8%~53.3%)."

42.     Lines 343-347: Numbers of OFP here are different than in the rest of the manuscript.

Response: Thank you for pointing out the discrepancy. We apologize for the inconsistency in the reported OFP values. After reviewing the calculations, we found that the OFP numbers in Lines 343-

347 in the original manuscript were incorrect. It appears that the necessary corrections were inadvertently omitted during the manuscript revision process. We have now corrected the OFP values in the current manuscript. Please refer to Lines 505-511:

"The OFP in Zibo is $410.4\pm197.2$ μg m$^{-3}$, with OVOCs accounting for the largest proportion (31.5%~55.9%). The upwind site (CQ, $464.2\pm162.3$ μg m$^{-3}$) has the highest OFP, followed by the suburban site (TZ, $456.3\pm295.3$ μg m$^{-3}$), the urban site (ZL, $441.1\pm174.5$ μg m$^{-3}$), the industrial site (XD, $422.9\pm166.9$ μg m$^{-3}$), and the downwind site (CD, $279.4\pm101.2$ μg m$^{-3}$) (Table S5). OFP contributed by OVOCs is most dominant at suburban (TZ, 254.9 μg m$^{-3}$) and industrial (XD, 194.7 μg m$^{-3}$) sites, followed by urban (ZL, 148.9 μg m$^{-3}$), upwind (CQ, 146.2 μg m$^{-3}$), and downwind (CD, 102.3 μg m$^{-3}$) sites."

Supplementary Information

43.     A lot of information is provided in the supplement only, but is discussed widely in the article. Please, reconsider using some of it for clarifications of the main text.

Response: Thank you for your valuable suggestion. We agree that certain supplementary information discussed in the main text could enhance the clarity and completeness of the article. We have carefully reviewed the supplementary content and integrate relevant details into the main text where appropriate to improve the flow and understanding for the readers.

---

## Author Response (AR2)

Dear Editor,

Many thanks for your suggestions. The title of the manuscript has been changed with "Significant influence of oxygenated volatile organic compounds on atmospheric chemistry: A case study in a typical industrial city in China". In addition, the ppb unit used in the previous manuscript has been replaced by $10^{-9}$ in the revised manuscript.

Once again, I would like to extend my heartfelt thanks to the you. Your meticulous work and valuable suggestions have significantly enhanced the quality of this piece. I truly appreciate your professionalism and dedication.

Best regards,

Kun